# Predicting integers from continuous parameters

**Bas Maat**°                                               *research@revstaudio.com*
*REVST*

**Peter Bloem**°                                            *contin@peterbloem.nl*
*Learning & Reasoning Group*
*Vrije Universiteit Amsterdam*

**Reviewed on OpenReview:** *https://openreview.net/forum?id=d1WKFlKFEa*

## Abstract

We study the problem of predicting numeric labels that are constrained to the integers or to a subrange of the integers. For example, the number of up-votes on social media posts, or the number of bicycles available at a public rental station. While it is possible to model these as continuous values, and to apply traditional regression, this approach changes the underlying distribution on the labels from discrete to continuous. Discrete distributions have certain benefits, which leads us to the question whether such integer labels can be modeled directly by a discrete distribution, whose parameters are predicted from the features of a given instance. Moreover, we focus on the use case of output distributions of neural networks, which adds the requirement that the *parameters* of the distribution be continuous so that backpropagation and gradient descent may be used to learn the weights of the network. We investigate several options for such distributions, some existing and some novel, and test them on a range of tasks, including tabular learning, sequential prediction and image generation. We find that overall the best performance comes from two distributions: *Bitwise*, which represents the target integer in bits and places a Bernoulli distribution on each, and a discrete analogue of the Laplace distribution, which uses a distribution with exponentially decaying tails around a continuous mean.

## 1 Introduction

In many regression tasks, the value to be predicted is discrete. One example is predicting the net migration between countries or cities. This can take any negative or positive value, but it must always be an integer. In other cases, the target value lies in a subset of the integers. A particularly common case is *count data*: predicting a number of items, like the number of bicycles available at a location for public bike rental service. We may even predict values in an interval, like the 256 values per color channel that an RGB pixel can take.

In machine learning, this problem is often solved by *continuous relaxation*: we simply interpret the label as a continuous value, train our model with a continuous loss like mean-squared error, and round the prediction to the nearest integer at inference time. While ad-hoc, this approach often works reasonably well. In some cases, however, like the pixel prediction example, such continuous relaxations fall far short of other approaches (Loaiza-Ganem & Cunningham, 2019; Rybkin et al., 2021).

A more principled approach is to define a probability distribution on the allowed target labels. We can then maximize the log-likelihood simply by taking as our loss the negative log-probability that the model assigns to the correct value. One simple example, often used in image generation (Rybkin et al., 2021; van den Oord et al., 2016), is to use a categorical distribution: the model produces a probability vector with one element for every allowed target value. This has two downsides. First, on large numbers of allowed values it quickly becomes impractical, and for the full range of integers and natural numbers it is impossible. Second,

---

° Shared first authors.

it ignores the ordinal nature of the target values: the outputs are modeled as a set and their ordering needs to be learned from the data.

Here, we investigate the feasibility of using genuine distributions on the integers—or subranges of the integers—for regression tasks. We investigate several existing candidates, and introduce three approaches that, while not entirely novel, have never been tested in this precise setting before.

Which brings us to our research question: For a task which includes integer labels, can we perform regression *without* re-interpreting the labels as continuous values? That is, can we devise a distribution on the integers, possibly constrained to a given range, which is suitable for the purposes of regression?

This is not a novel problem, and some approaches exist. However, we will focus on regression models that are optimized by backpropagation and gradient descent: i.e. *neural networks*. This gives us three additional requirements:

1. While the sample space of the distribution (the integers) is *discrete*, its parameters should be *continuous*. This will allow us to define a gradient of the negative log-probability with respect to the parameters, which we can then backpropagate through whatever mechanism produced the parameters.

2. The gradient should be *well-behaved*. It should, ideally, be defined everywhere, and not come too near 0 at any point in the parameter space, or have too many discontinuities.

3. The distribution should be able to "peak" around an arbitrary point. That is, if we make a prediction $x$ about which we are very certain, then we should be able to set the parameters in such a way that almost all probability is assigned to $x$.[1]

Such a distribution would have many potential benefits. Foremost, it provides a kind of conceptual simplicity. To illustrate, consider the units of the loss function. For a discrete distribution, the log-probability of event $x$ is a value in bits. It expresses the bits required to communicate $x$, under an optimal encoding for the distribution (Grünwald, 2007, Section 3.2). This scale has a meaningful minimum at 0: with $p(x) = 1$, we can communicate $x$ with 0 bits, which is the best we can do.

For continuous distributions, this interpretation breaks down: the log-probability *density* does not have any units with a natural interpretation and is not restricted to positive values. Moreover, it is not invariant to a change of variables (Cover, 1999, Theorem 8.6.4): we get different values depending on which coordinates we use for the target variable.

This conceptual elegance has practical benefits. Consider a setting where we must predict multiple values, say a flight simulator where we choose our engine thrust from an integer range and our flight stick position from the unordered set {up, down, left, right, neutral}. Modeling both values as probabilities results in log-losses that can be neatly and simply summed together. While it's not guaranteed that the losses will balance, adding them together without a multiplier gives us a reasonable place to start, since the magnitude of the two losses is proportional to the representational complexity of the two values. That is, we can describe the current state of the controls in $-\log p(\text{thrust}) - \log p(\text{stick})$ bits, so summing the two log-losses together is a natural way to balance them.

Compare this with the case where we model the thrust with a continuous relaxation. In that case we get a value of arbitrary magnitude for the thrust's log-density, which could even be negative. When we add the log probability density to the log-probability for the stick position, we are forced to add a multiplier, which must be tuned. Moreover the sum of the two losses is no longer an interpretable value.

This also has implications for situations where we predict a single value, since we often want to compare losses between models. With log-probability densities, this is not an option unless the models are guaranteed to use the same distribution with the same parameterization. With log-probabilities, even models using categorical distributions can be compared to models using ordinal distributions. This is especially apparent

---

[1]This rules out, for example, the Poisson distribution. Its parameter $\lambda$ is continuous and its sample space discrete, but the probability mass cannot be made arbitrarily concentrated on a given integer.

in generative modeling. Model comparison is a challenging task, with authors often relying on ad-hoc measures like inception scores. However, when the model produces probabilities for a given instance, the bits per instance can be computed and compared between radically different models. This has provided strong signals in domains like text generation (Radford et al., 2019) and image generation (Ho et al., 2020).

In short, discrete probability distributions are promising tools if they can be made to work as the output distributions of neural nets. The categorical distribution, in the form of a softmaxed output vector is a tried-and-tested candidate. However, it (a) only works for finite ranges (b) is parameter inefficient for large ranges (c) doesn't model the ordinality of the sample space. To achieve the same benefits for unbounded integer ranges, with efficient use of parameters and ordinality hardcoded, we need integer distributions with the properties given above.

We identify three existing candidates to be used as baselines and introduce three more novel approaches.[2]

In order, these are:

**Discrete Weibull** A discrete analogue of the Weibull distribution.

**Discretized Normal** The normal distribution, discretized by assigning the probability mass on the interval $[x - \frac{1}{2}, x + \frac{1}{2})$ to the integer $x$.

**Discretized Laplace** The Laplace distribution, discretized in the same way.

**Dalap** Short for *discrete analogue to the Laplace distribution*. Takes the form $p(n) \propto \gamma^{|n-\mu|}$. This distribution was introduced with a discrete parameter $\mu$ in (Inusah & Kozubowski, 2006). We extend this to a continuous parameter, and derive the partition function.

**Danorm** Short for *discrete analogue to the Normal distribution*. Takes the form $p(n) \propto \gamma^{(n-\mu)^2}$. Introduced in a different form in (Kemp, 1997). We use the form above, by analogy with Dalap, and approximate the partition function.

**Bitwise** This distribution is based on the common idea of representing an integer as a bitstring, in sign-magnitude representation, and having a model produce a Bernoulli distribution for each bit. While this representation is common in neural networks, we have not found references that interpret the resulting model as a distribution on the integers.

We study these distributions in three settings. First, we define three tabular regression tasks with integer labels. These are based on the number of bicycles available at public rental stations, the number of upvotes on content in a social media setting and the net migration between countries. Next, we test the models in a sequential prediction setting. Specifically a MIDI music prediction task. We isolate the issue of predicting the ticks on the following note, given the sequence of events that came before. Finally, we show a proof-of-concept for integer prediction in the image generation domain. We train an autoregressive pixel model on images, treating each channel of each pixel as an integer prediction problem restricted to the range $[0, 255]$.

Our findings are that Dalap outperforms the existing distributions in most cases when performance is measured in negative log-probability. For point-prediction accuracy, continuous squared-error regression remains a strong baseline, although mixture models with discrete distributions can outperform it on some datasets. Danorm provides a natural squared-error-like discrete loss, but its rapidly decaying tails often lead to substantially worse negative log-probability, and its numerical partition function makes it impractical for the image experiments.

In most tabular data, relaxing the problem to continuous target labels leads to a better RMSE score than any of the discrete distributions, suggesting that in most settings, that is likely still a competitive approach if performance outweighs conceptual neatness or interpretability. However, when a discrete distribution is required, Bitwise, Dalap and Danorm are likely the best options.

---

[2]Code is available at `https://github.com/R3VST/predicting-integers-from-continuous-parameters`.

## 2 Related Work

We briefly discuss the large variety of models for count data (i.e. those supported on $\mathbb{N}$) and for integer data (supported on $\mathbb{Z}$). Most of these do not fit our requirements outlined in the introduction. Others have had to be excluded to keep the scope of the present research manageable.

### 2.1 Count models

A subset of regression on integer-valued labels is that of modeling *count data*, where the labels are restricted to the non-negative or positive integers and usually represent counts of items or events. This problem is well-studied and is often tackled with Poisson or negative binomial distributions (Cameron & Trivedi, 2013; Hilbe, 2014).

While our approach shares a lot with this existing line of research, the focus on neural network modeling changes the problem in some significant ways. First, we are not interested in closed-form solutions, or convergent optimization for the optimal parameters, since these properties are not available for the weights of the neural network in any case. Second, in our setting the parameters of the distribution are required to be continuous, so that a gradient can be defined for them. And, finally, we are only interested in conditional modeling: predicting integer labels given some features or other conditioning information, rather than fitting a single distribution to a whole population.

**Poisson-based**  A very common family of approaches is built on the Poisson distribution. Since this distribution has only one parameter, the mean and variance must be related. In fact in the Poisson distribution, they are equal. This is referred to as equidispersion. Various extensions have been proposed (Bonat et al., 2018) to allow for overdispersion (a variance higher than the mean) and underdispersion (a variance lower than the mean). However, this still puts a lower or upper bound on the variance that depends on the chosen mean. For our purposes, we require a distribution where the variance can be set independently from the mean, as it is in, for example, the normal distribution. Ideally, the distribution can be parameterized directly in terms of the required location and dispersion. Later models in this family offer great flexibility in the dispersion (Shmueli et al., 2005), but this often comes at the cost of difficult to estimate probabilities. For these reasons we consider this family as a whole out of scope.

**Discrete Weibull**  One prominent example of a discrete distribution with continuous parameters, is the *Discrete Weibull distribution* (Nakagawa & Osaki, 1975; Collins et al., 2020; Englehardt & Li, 2011). We include this as a baseline and describe it briefly in the next section. Generalizations include the exponentiated discrete Weibull distribution (Nekoukhou & Bidram, 2015), which can take bimodal ("bathtub") shapes as well as unimodal. For our purposes, this is not required behavior, so we consider generalizations out of scope.

**Geometric distributions**  As the tails of Dalap follow a geometric progression, it is related to the geometric distribution. Some other discrete generalizations of the geometric distribution exist. One example is (Gómez-Déniz, 2010), which allows the mode to take values other than 0. However, in this distribution, the mean is not a straightforward function of the parameters, and the mean and dispersion are not independent, which are important properties for our use case.

**Others**  There are other discrete distributions, which due to their shape, we did not consider promising candidates. These include the discrete Lindley distribution (Gómez-Déniz & Calderín-Ojeda, 2011), the discrete Rayleigh distribution (Roy, 2004) and the discrete Burr-Hatke distribution (El-Morshedy et al., 2020). In general these models are likely to show overdispersion in a neural network setting. That is, the data distribution (conditioned on a particular set of feature-values) is likely to form a low-variance, high peaked distribution around a single value, which the distribution cannot represent.

## 2.2 Integer models

The broader problem of *integer regression*, where the target label is a discrete, but potentially negative value is less recognised in the literature. We expect that this is because the continuous relaxation poses fewer problems in these domains than it does for count data.

### 2.2.1 Discretization of continuous distributions

A straightforward approach to building a distribution on the integers is to start with a continuous distribution, like a Normal distribution, and to discretize a sample from it. We include two such discretizations below, using a simple rounding operator. In (Tovissodé et al., 2021) a probabilistic discretization scheme is introduced. This scheme has the benefit over the rounding approach that it preserves expectations.

A slightly different approach is that of creating discrete *analogues* (Alzaatreh et al., 2012). Dalap and Danorm are instances of this approach. In these cases one proceeds by taking key attributes of a continuous distribution, like a maximum entropy property, and finding the integer distribution that shares these properties. In some cases, like the discretized Normal distribution, both approaches yield the same result, but in others, like the discretized Laplace, different solutions may emerge. For a general survey of discretization and discrete analogues, we refer the reader to (Chakraborty, 2015).

## 2.3 Other approaches

Cameron and Johansson (Cameron & Johansson, 1997) introduce a method to modify the density of an arbitrary discrete distribution, with parameters $\theta$ by a polynomial with parameters $\mathbf{a}$. The parameters $\mathbf{a}$ modify the moments of the base distribution in a predictable way, allowing for over- and underdispersion, and the derivative of the log-likelihood with respect to $\theta$ and $\mathbf{a}$ is readily available. We treat this as an extension that may be applied to all integer distributions, including our baselines, and our newly introduced approaches, as well as distributions like the Poisson, which were previously ruled out as viable candidates.

# 3 Methods

In this section, we briefly detail the three existing distributions that satisfy our requirements (discretized Normal, discretized Laplace and Weibull), and then present our three partly novel approaches (Dalap, Danorm and Bitwise).

## 3.1 Discretized Normal (`dnormal`)

As noted in the related work, there are various ways to discretize the normal distribution to the integers. For our purposes, the most direct approach is simply to sample from a normal distribution and then to round the output to the nearest integer.

More formally, let $z \sim N(\mu, \sigma)$ with some mean $\mu$ and standard deviation $\sigma$. Let $r : \mathbb{R} \to \mathbb{Z}$ be the rounding function which maps the real values $[n - \frac{1}{2}, n + \frac{1}{2})$ to the integer value $n$. Then, we call the distribution on $r(z)$ the *discretized normal distribution* $N'$ with parameters $\mu, \sigma$.

The probability mass $N'(n \mid \mu, \sigma^2)$ can be computed simply from the cumulative distribution function of $N$:

$$N'(n \mid \mu, \sigma^2) = N\left(n - \frac{1}{2} \leq z < n + \frac{1}{2} \mid \mu, \sigma\right)$$
$$= N\left(z < n + \frac{1}{2} \mid \mu, \sigma\right) - N\left(z < n - \frac{1}{2} \mid \mu, \sigma\right).$$

The cumulative distribution can be written in terms of the *error function* as follows.

$$N(z < b \mid \mu, \sigma) = \frac{1}{2}\left[1 + \mathrm{erf}\left(\frac{b - \mu}{\sigma\sqrt{2} + \epsilon}\right)\right]$$

with

$$\text{erf}(x) = \frac{2}{\sqrt{\pi}} \int_0^x e^{-t^2} \, dt.$$

While the error function has no closed-form solution, various approximations are available which are closed-form.[3] For the backward pass, the derivative

$$\frac{\partial \text{erf}(x)}{\partial x} = \frac{2}{\sqrt{\pi}} e^{-x^2}$$

is easily computed.

## 3.2 Discretized Laplace (`dlaplace`)

We discretize the Laplace distribution in the same manner as the normal distribution. Let

$$z \sim \text{Laplace}(\mu, b)$$

be a continuous Laplace random variable with location parameter $\mu$ and scale parameter $b$. Its probability density function is given by

$$f(z \mid \mu, b) = \frac{1}{2b} \exp\left(-\frac{|z - \mu|}{b}\right).$$

Let $F(z \mid \mu, b)$ denote the cumulative distribution function of the Laplace distribution:

$$F(z \mid \mu, b) = \begin{cases} \frac{1}{2} \exp\left(\frac{z-\mu}{b}\right) & \text{if } z < \mu, \\ 1 - \frac{1}{2} \exp\left(-\frac{z-\mu}{b}\right) & \text{if } z \geq \mu. \end{cases}$$

Let $r : \mathbb{R} \to \mathbb{Z}$ be defined as in Section 3.1 We then call the distribution of $r(z)$ the *discretized Laplace distribution.*

The probability mass assigned to an integer $n$ is computed as

$$P(n \mid \mu, b) = F\left(n + \frac{1}{2} \mid \mu, b\right) - F\left(n - \frac{1}{2} \mid \mu, b\right).$$

## 3.3 Discrete Weibull

The discrete Weibull distribution is an extension of the Weibull distribution Nakagawa & Osaki (1975), specifically designed for discrete, non-negative data, making it well-suited for modeling count data. Unlike the continuous Weibull distribution, the discrete version is characterized by its ability to model varying degrees of dispersion in count data. The probability mass function (PMF) of the discrete Weibull distribution is given by equation 1, where $\alpha > 0$ is the scale parameter and $\beta > 0$ is the shape parameter. This difference between consecutive values ensures that each integer count $x$ has a defined probability.

$$p(x) = \exp\left(-\left(\frac{x}{\alpha}\right)^\beta\right) - \exp\left(-\left(\frac{x+1}{\alpha}\right)^\beta\right) \tag{1}$$

Which leads to the following log-probability.

$$\log p(x) = \log\left(1 - \exp\left(\left(\frac{x}{\alpha}\right)^\beta - \left(\frac{x+1}{\alpha}\right)^\beta\right)\right) - \left(\frac{x}{\alpha}\right)^\beta$$

---

[3]In our experiments, we use the Pytorch implementation of the error function. This defers to various closed-source systems (depending on backend), for which the approximation used is not usually documented. We use the CUDA backend in all experiments.

This implementation of the log-probability may lead to unstable gradients. In our implementation, we force the log-probability to underflow to negative infinity in case both $-\left(\frac{x}{\alpha}\right)^\beta$ and $-\left(\frac{x+1}{\alpha}\right)^\beta$ go to negative infinity. For details, see Section A.5.1.

### 3.4 Discrete analogue of the Laplace distribution (`dalap`)

In Inusah & Kozubowski (2006), the authors introduce the following discrete analogue of the Laplace distribution:

$$p(n \mid \mu, \gamma) = \frac{1 - \gamma}{1 + \gamma} \gamma^{|n - \mu|}. \tag{2}$$

Here, $\gamma$ is a continuous value between 0 and 1, and $\mu$ and $n$ are integers. For our purposes, this is not yet a suitable distribution, because $\mu$ is not continuous.

A simple approach is to assume that $\mu$ is continuous, and to let $p(n \mid \mu, \gamma) \propto \gamma^{|n - \mu|}$, where $n - \mu$ is now a continuous value. In order to achieve this, we must change the partition function. First, let

$$f = \mu - \lfloor \mu \rfloor \qquad\qquad c = \lceil \mu \rceil - \mu \qquad\qquad \text{dist. to the nearest neighbor of } \mu \ [4]$$

So that we have

$$p(n \mid \mu, \gamma) \propto \begin{cases} \gamma^f \gamma^{\lfloor \mu \rfloor - n} & \text{if } n < \mu \\ \gamma^c \gamma^{n - \lceil \mu \rceil} & \text{if } n \geq \mu \end{cases}$$

Then, for the sum of the unnormalized probability mass $z$, we have

$$\begin{aligned}
z = \sum_{i \in \mathbb{Z}} \gamma^{|i - \mu|} &= \sum_{i = -\infty}^{\infty} \begin{cases} \gamma^f \gamma^{\lfloor \mu \rfloor - i} & \text{if } i \leq \mu \\ \gamma^c \gamma^{i - \lceil \mu \rceil} & \text{if } i > \mu \end{cases} \\
&= \gamma^f \sum_{i <= \mu} \gamma^{\lfloor \mu \rfloor - i} + \gamma^c \sum_{i > \mu} \gamma^{i - \lceil \mu \rceil} \\
&= \gamma^f \sum_{j=0}^{\infty} \gamma^j + \gamma^c \sum_{j=0}^{\infty} \gamma^j = (\gamma^c + \gamma^f) \frac{1}{1 - \gamma}.
\end{aligned}$$

Where the last step uses the well known relation $\sum_{i=0}^{\infty} \alpha^i = \frac{1}{1-\alpha}$.

Using $z^{-1}$ as a partition function, we get

$$p(n \mid \mu, \gamma) = \frac{1 - \gamma}{\gamma^{\mu - \lfloor \mu \rfloor} + \gamma^{\lceil \mu \rceil - \mu}} \gamma^{|n - \mu|}$$

which reduces to (2) in cases where $\mu$ takes an integer value, (as the denominator reduces to $\gamma^0 + \gamma^1 = 1 + \gamma$).

Note that even in cases where $\mu$ is just below or just above an integer value, this definition ensures that a small change in $\mu$ results in a small change to $p$, as required for effective use in gradient descent.

---

[4]We define $\lceil \mu \rceil = \lfloor \mu \rfloor + 1$ to avoid instabilities when $\mu$ takes an integer value.

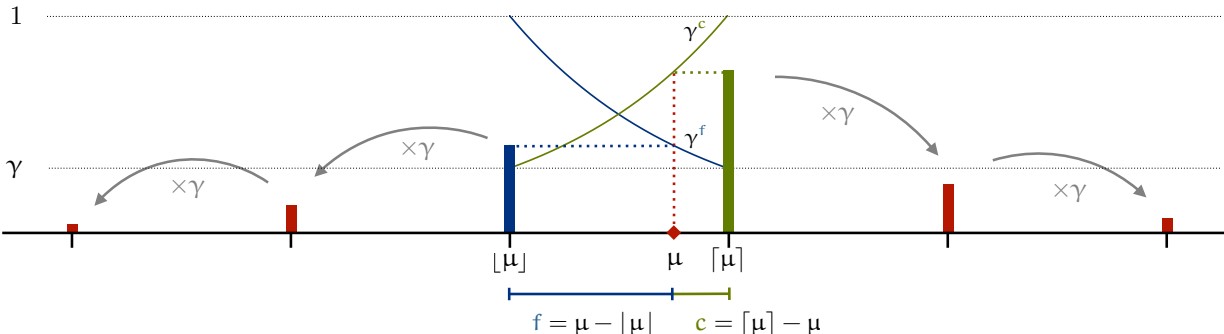

Figure 1: The principle behind Dalap. The neighbors of $\mu$ are assigned probability mass between $\gamma$ and 1 by an exponential function of their distance to $\mu$. The rest of the distribution decays geometrically from these two values.

### 3.5 Bounded versions

To constrain the distribution to a subset $S$ of $\mathbb{Z}$ we take the unconstrained version of $p$ and condition on a subset of $\mathbb{Z}$. In such cases, the probability mass function is the same as above, except for $z$.[5]

We specify two cases: on a one-way bounded interval $[l, \infty)$, with $\mathbb{N}$ as a special case, and on a two-way bounded interval $[l, u]$.

For the first, assume first that the interval is $[0, \infty)$, from which the general case will follow by change of variables. We have

$$
\begin{aligned}
z_1 &= \sum_{n=0}^{\infty} \begin{cases} \gamma^f \gamma^{\lfloor \mu \rfloor - n} & \text{if } n \leq \lfloor \mu \rfloor \\ \gamma^c \gamma^{n - \lceil \mu \rceil} & \text{if } n \geq \lceil \mu \rceil \end{cases} \\
&= \gamma_f \sum_{n=0}^{\lfloor \mu \rfloor} \gamma^{\lfloor \mu \rfloor - n} + \gamma_c \sum_{n=\lceil \mu \rceil}^{\infty} \gamma^{n - \lceil \mu \rceil} \\
&= \gamma^f \sum_{i=0}^{\lfloor \mu \rfloor} \gamma^i + \gamma^c \sum_{i=0}^{\infty} \gamma^i \\
&= \gamma^f \frac{1 - \gamma^{\lfloor \mu \rfloor + 1}}{1 - \gamma} + \gamma^c \frac{1}{1 - \gamma}. \qquad \text{\textcolor{red}{note that } } \sum_{i=0}^{n} \alpha^i = \frac{1 - \alpha^{n+1}}{1 - \alpha} \\
&= \frac{\gamma^{\mu - \lfloor \mu \rfloor}(1 - \gamma^{\lfloor \mu \rfloor + 1})}{1 - \gamma} + \frac{\gamma^{\lceil \mu \rceil - \mu}}{1 - \gamma}. \\
&= \frac{\gamma^{\mu - \lfloor \mu \rfloor} - \gamma^{\lfloor \mu \rfloor + 1 + \mu - \lfloor \mu \rfloor} + \gamma^{\lceil \mu \rceil - \mu}}{1 - \gamma}. \\
&= \frac{\gamma^{\mu - \lfloor \mu \rfloor} - \gamma^{\mu + 1} + \gamma^{\lceil \mu \rceil - \mu}}{1 - \gamma}.
\end{aligned}
$$

Which gives us

---

[5]By the definition of conditional probability

$$
p(n \mid n \in S) = \frac{p(n)}{\sum_{m \in S} p(m)} = \frac{z^{-1} q(n)}{\sum_m z^{-1} q(m)} = \frac{1}{\sum_m q(m)} q(n)
$$

.

$$p_{\mu,\gamma}(n \mid n \in \mathbb{N}) = \frac{1 - \gamma}{\gamma^{\mu - \lfloor \mu \rfloor} + \gamma^{\lceil \mu \rceil - \mu} - \gamma^{\mu + 1}} \gamma^{|n - \mu|}.$$

For other bounds $[l, \infty)$, we reparametrize with[6]

$$p_{\mu,\gamma}(n \mid n \geq l) = p_{\mu + l, \gamma}(n + l \mid n \in \mathbb{N}).$$

Finally, for a two-way bounded interval $[l, u]$, we have

$$
\begin{aligned}
z_2 &= \sum_{n=l}^{u} \begin{cases} \gamma^f \gamma^{\lfloor \mu \rfloor - n} & \text{if } n \leq \lfloor \mu \rfloor \\ \gamma^c \gamma^{n - \lceil \mu \rceil} & \text{if } n \geq \lceil \mu \rceil \end{cases} \\
&= \gamma^f \sum_{n=l}^{\lfloor \mu \rfloor} \gamma^{\lfloor \mu \rfloor - n} + \gamma^c \sum_{n=\lceil \mu \rceil}^{u} \gamma^{n - \lceil \mu \rceil} \\
&= \gamma^f \sum_{i=0}^{\lfloor \mu \rfloor - l} \gamma^i + \gamma^c \sum_{i=0}^{u - \lceil \mu \rceil} \gamma^i \\
&= \gamma^f \frac{1 - \gamma^{\lfloor \mu \rfloor - l + 1}}{1 - \gamma} + \gamma^c \frac{1 - \gamma^{u - \lceil \mu \rceil + 1}}{1 - \gamma} \\
&= \frac{\gamma^{\mu - \lfloor \mu \rfloor}(1 - \gamma^{\lfloor \mu \rfloor - l + 1})}{1 - \gamma} + \frac{\gamma^{\lceil \mu \rceil - \mu}(1 - \gamma^{u - \lceil \mu \rceil + 1})}{1 - \gamma} \\
&= \frac{\gamma^{\mu - \lfloor \mu \rfloor} - \gamma^{\lfloor \mu \rfloor - l + 1 + \mu - \lfloor \mu \rfloor} + \gamma^{\lceil \mu \rceil - \mu} - \gamma^{u - \lceil \mu \rceil + 1 + \lceil \mu \rceil - \mu}}{1 - \gamma} \\
&= \frac{\gamma^{\mu - \lfloor \mu \rfloor} + \gamma^{\lceil \mu \rceil - \mu} - \gamma^{1 + \mu - l} - \gamma^{1 - \mu + u}}{1 - \gamma}
\end{aligned}
$$

Which gives us

$$p_{\mu,\gamma}(n \mid n \in [l, u]) = \frac{1 - \gamma}{\gamma^{\mu - \lfloor \mu \rfloor} + \gamma^{\lceil \mu \rceil - \mu} - \gamma^{1 + \mu - l} - \gamma^{1 - \mu + u}} \gamma^{|n - \mu|}.$$

The negative log probabilities for all three versions simplify to

$$-\log p_{\mu,\gamma}(n) = |\mu - n| \log \frac{1}{\gamma} + \log z$$

where $z$ depends only on $\gamma$ in the unbounded case, and on both $\mu$ and $\gamma$ in the bounded cases.

Note that this preserves a key property of the non-discrete Laplace distribution that we did not see in the simpler discretized Laplace above: the loss is a linear function of the error $|\mu - n|$ (with $\log \frac{1}{\gamma}$ a positive value, since $\gamma < 1$). That is, to minimize the loss, we minimize the absolute distance between the model prediction $\mu$ and the target value $n$. This loss is then tempered by the amount of uncertainty in a multiplicative factor $\log \frac{1}{\gamma}$ and a regularization term $\log z$.

See Appendix A.3 for a further analysis of the terms of $-\log p(n)$. Implementation details for numerical stability of the bounded partition function are provided in Appendix A.5.

---

[6]In the rare case of an interval bounded from above, we can reparametrize by first multiplying by $-1$.

### 3.6 Mean and variance.

We will finish up this section with a brief analysis of the mean and variance of the Dalap distribution. For its intended purpose as a neural network loss function, we primarily want to show that the variance can be made sufficiently small around a given expected value, and that this expected value is a simple function of the parameters. In the case of Dalap, we can show that the parameter $\mu$, while not equal to the expected value in general, does indeed coincide with the expected value as the variance goes to zero, which we accomplish by letting $\gamma$ go to zero.

**Proposition 1.** *In the unbounded case, the expected value of $p(n \mid \mu, \gamma)$ is*

$$\mathcal{E}_{\mu,\gamma} n = \frac{\gamma^f}{\gamma^f + \gamma^c} \left( \lfloor \mu \rfloor - \frac{\gamma}{1-\gamma} \right) + \frac{\gamma^c}{\gamma^f + \gamma^c} \left( \lceil \mu \rceil + \frac{\gamma}{1-\gamma} \right)$$

*Proof.*

$$\mathcal{E}_{\mu,\gamma} n = \sum_{n \in \mathbb{Z}} p(n \mid \mu, \gamma) n = \frac{1}{z} \gamma^f \sum_{i=0}^{\infty} \gamma^i (\lfloor \mu \rfloor - i) + \frac{1}{z} \gamma^c \sum_{i=0}^{\infty} \gamma^i (\lceil \mu \rceil + i)$$

$$= \gamma^f \lfloor \mu \rfloor \frac{1}{z} \sum_i \gamma^i - \gamma^f \frac{1}{z} \sum_i \gamma^i i + \gamma^c \lceil \mu \rceil \frac{1}{z} \sum_i \gamma^i + \gamma^c \frac{1}{z} \sum_i \gamma^i i$$

$$= (\gamma^f \lfloor \mu \rfloor + \gamma^c \lceil \mu \rceil) \frac{1}{z} \frac{1}{1-\gamma} + (\gamma^c - \gamma^f) \frac{1}{z} \frac{\gamma}{(1-\gamma)^2} \qquad \text{note } \sum_i \alpha^i i = \alpha/(1-\alpha)^2$$

$$= \frac{1}{z} \frac{\gamma^f \lfloor \mu \rfloor + \gamma^c \lceil \mu \rceil + \gamma^c \frac{\gamma}{1-\gamma} - \gamma^f \frac{\gamma}{1-\gamma}}{1-\gamma}$$

$$= \frac{1-\gamma}{\gamma^f + \gamma^c} \frac{\gamma^f (\lfloor \mu \rfloor - \frac{\gamma}{1-\gamma}) + \gamma^c (\lceil \mu \rceil + \frac{\gamma}{1-\gamma})}{1-\gamma}$$

$$= \frac{\gamma^f \left( \lfloor \mu \rfloor - \frac{\gamma}{1-\gamma} \right) + \gamma^c \left( \lceil \mu \rceil + \frac{\gamma}{1-\gamma} \right)}{\gamma^f + \gamma^c}$$

From which the result follows by re-arrangement. □

This is a weighted sum between a point some distance $\frac{1}{1-\gamma}$ below $\lfloor \mu \rfloor$ and another point the same distance above $\lceil \mu \rceil$. The point $\lfloor \mu \rfloor - \frac{\gamma}{1-\gamma}$ is the center of mass of the left tail, if $f = 1$ and likewise for the other term and the right tail. The expectation is a convex combination of both points.

The weights are how much $\gamma^f$ and $\gamma^c$ respectively claim of the sum $\gamma^f + \gamma^c$. See Figure 2 for an illustration.

To understand the behavior of this function, first set $\mu$ equal to an integer value. Then, $f$ and $c$ go to $\gamma$ and 1. If we then let $\gamma$ go to 0, we see that the weights go to 0 and 1 and only one of the terms (corresponding to $\mu$) remains. In this term, the distance goes to zero with $\gamma$ and we are left with the value $\mu$.

If $\mu$ does not take an integer value, and we let $\gamma$ go to zero, we find

$$\lim_{\gamma \to 0} \frac{\gamma^f}{\gamma^f + \gamma^c} = \lim_{\gamma \to 0} \frac{1}{1 + \gamma^{c-f}} = \begin{cases} 1 & \text{if } c > f \\ 1/2 & \text{if } c = f \\ 0 & \text{if } c < f \end{cases} \tag{3}$$

and likewise with $f$ and $c$ reversed. This tells us that in the limit, the expectation becomes equal to the nearest integer to $\mu$ (that is, $r(\mu)$) unless $\mu$ is precisely between two integers, in which case the expectation is $\mu$.

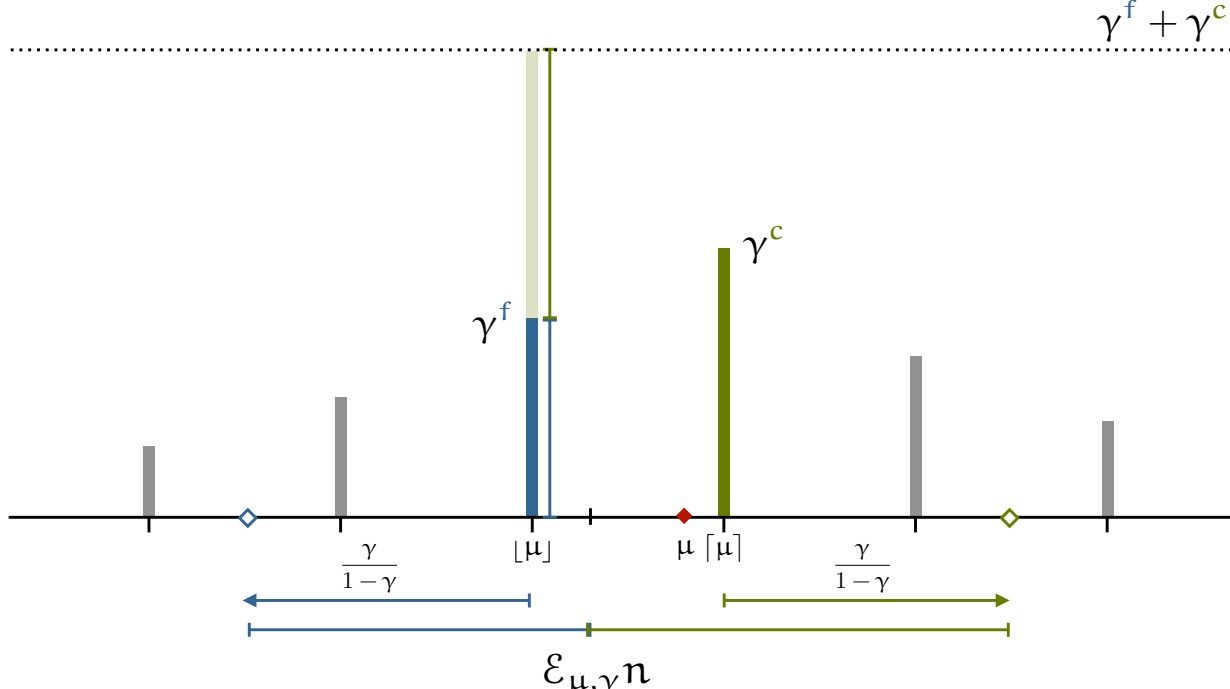

Figure 2: The expected value of unbounded Dalap. We take two numbers at a distance of $\frac{\gamma}{1-\gamma}$ below and above $\lfloor\mu\rfloor$ and $\lceil\mu\rceil$ respectively. The expected value is a weighted mean of these with weights proportional to $\gamma^f$ and $\gamma^c$.

This shows an inductive bias for integer values: as we move to absolute certainty ($\gamma \to 0$), the expectation takes integer values (except for a zero-measure set of values of $\mu$ which leads to half-integers).

For the bounded versions of Dalap, we won't derive the full variance, but we will show that the value converges to $r(\mu)$ as $\gamma \to 0$ in the unbounded case, so long as the two neighboring integers are in the support.

**Proposition 2.** *If $\lfloor\mu\rfloor, \lceil\mu\rceil \in S$ then $\mathcal{E}_{n\sim p(n|n\in S,\mu,\gamma)}n$ goes to $r(\mu)$ as $\gamma \to 0$ unless $c = f$ in which case it goes to $\mu$.*

*Proof.* If $\mu$ is an integer, the probabilities of $\mu - 1$ and $\mu + 1$ are $\gamma p(\mu)$ and all other integers have lower probability. As $\gamma \to 0$, these go to 0, suggesting that all probability mass is assigned to $\mu$, proving the proposition.

For the remainder, assume that $\mu$ is not an integer. We first write $\mathcal{E}n = \sum_{n\in S} p(n)n$.

The probabilities in this sum are of the form $\gamma^c\gamma^i/(\sum_j \gamma^f\gamma^j + \sum_j \gamma^c\gamma^j)$ for the right tail (with $c$ replaced by $f$ in the numerator for the left tail). Writing

$$\lim_{\gamma\to 0} \frac{\gamma^c\gamma^i}{(\gamma^f + \gamma^c)\sum_j \gamma^j} = \lim_{\gamma\to 0} \gamma^i \frac{\gamma^c}{\gamma^f + \gamma^c}(1-\gamma)$$

We note that the factor $\gamma^i$ causes the probability to go to zero, unless $i = 0$. This is only the case for the two neighbors of $\mu$, for which the limit goes to $\frac{\gamma^c}{\gamma^f+\gamma^c}$ for $\lceil\mu\rceil$ and to $\frac{\gamma^f}{\gamma^f+\gamma^c}$ for $\lfloor\mu\rfloor$.

This gives us

$$\lim_{\gamma \to 0} \mathcal{E}n = \frac{\gamma^f}{\gamma^f + \gamma^c} \lfloor \mu \rfloor + \frac{\gamma^c}{\gamma^f + \gamma^c} \lceil \mu \rceil.$$

As shown in (3) if $c \neq f$, the probabilities go to 0 and 1 leaving only the term corresponding to $r(\mu)$. If $c = f$ the probabilities go to $\frac{1}{2}$, leaving us with the midpoint between the two integers which is (in this case) equal to $\mu$. $\qquad \square$

**Proposition 3.** *The variance of Dalap goes to 0 as $\gamma \to 0$, unless $c = f$ in which case it goes to $1/4$.*

*Proof.* Let $S \subseteq \mathbb{Z}$ be our support. We write the variance as

$$\mathcal{E}_n n^2 - (\mathcal{E}_n n)^2 = \sum_{n \in S} p(n) n^2 - (\mathcal{E}_n n)^2$$

Assume first that $c \neq f$. As $\gamma \to 0$, the second term goes to $r(\mu)^2$, following Proposition 2. If $\mu$ is an integer, we obtain $\mu^2 - r(\mu)^2 = 0$.

If $\mu$ is non-integer, only the direct neighbors of $\mu$ have nonzero probability in the limit (as shown in the proof of Proposition 2). For $c \neq f$, this leaves us with

$$p(\lfloor \mu \rfloor) \lfloor \mu \rfloor^2 + p(\lceil \mu \rceil) \lceil \mu \rceil^2 - (\mathcal{E}n)^2 = \frac{\gamma^f}{\gamma^f + \gamma^c} \lfloor \mu \rfloor^2 + \frac{\gamma^c}{\gamma^f + \gamma^c} \lceil \mu \rceil^2 - r(\mu)^2$$

Again, the probabilities go to 1 for the term corresponding to $r(\mu)$ and 0 for the other, leaving us with $r(\mu)^2 - r(\mu)^2 = 0$.

If $c = f$, $(\mathcal{E}n)^2$ goes to $\mu^2$ and the probabilities go to $1/2$, giving us

$$\mathcal{E}_n n^2 - (\mathcal{E}_n n)^2 = \frac{1}{2} \lfloor \mu \rfloor^2 + \frac{1}{2} \lceil \mu \rceil^2 - \mu^2$$

$$= \frac{\lfloor \mu \rfloor^2 + \lceil \mu \rceil^2}{2} - \left( \frac{\lfloor \mu \rfloor + \lceil \mu \rceil}{2} \right)^2 = \frac{1}{4}$$

Where the last step is derived by substituting $\lceil \mu \rceil = \lfloor \mu \rfloor + 1$ and multiplying out all squares. $\qquad \square$

This shows that the variance may be made arbitrarily small except for a negligible set of values of $\mu$, for which the variance is nevertheless a small non-zero value.

### 3.7 Discrete analogue of the normal distribution (`danorm`)

The main feature of the Laplace distribution is the exponential decay of its tails. Likewise, a defining feature of the normal distribution is the *squared*-exponential decay of its tails. Following this logic, we may ask whether we can take the exponential decay of Dalap and turn it into a squared exponential decay to obtain a discrete analogue for the normal distribution. This gives us the distribution

$$p(n \mid \mu, \gamma) \propto \gamma^{(n-\mu)^2}.$$

If we set $\gamma = e^{-\frac{1}{2s}}$—with $s$ some alternative dispersion parameter analogous to $\sigma^2$—we obtain the more familiar functional form of the normal distribution. Additionally, we can choose our parameters so that this distribution corresponds to the discrete analog of the normal distribution introduced in Kemp (1997). Like

the normal distribution on the real values, this is the maximum entropy distribution on the integers with given expectation and variance.

Assuming some partition function $z^{-1}$ and taking the negative log probability, we get

$$-\log p(n \mid \gamma, \mu) = (n - \mu)^2 \log \frac{1}{\gamma} + \log z.$$

This shows that, as it does for the normal distribution, our loss becomes a simple tempered version of the squared distance between our location parameter $\mu$ and the target value $n$. The tempering effect of $\log \frac{1}{\gamma}$ and $\log z$ is similar to that in Dalap, due to the similar functional forms.

This suggests that this distribution, which we'll call *Danorm*, is a natural choice for settings where the squared distance is what we are most interested in minimizing, but we still require a discrete distribution on the integers.

The downside of Danorm compared to Dalap is that the partition function has no closed form solution, because unlike the geometric tails of Dalap, there is no elementary closed-form for $\sum_i \gamma^{(i^2)}$.

Instead, we rely on the fast decay of the tails to compute $z$ numerically. For some parameter $n$ (500 in all experiments), we compute

$$\tilde{z} = \sum_{i \in [\lfloor \mu \rfloor - n, \lceil \mu \rceil + n]} \gamma^{(i - \mu)^2}$$

and substitute this for $z$. To ensure fast convergence in the tails, $\gamma$ can be activated to some maximal value slightly below 1.0.

For bounded versions of Danorm, we simply set out-of-bounds values to zero in this sum.[7]

This means that the time and memory complexity of computing the loss function increases by a factor of $2n$ compared to Dalap. In tabular settings, where we compute one probability per instance, this is unlikely to be a bottleneck. However, for something like high-resolution image generation, the overhead may be substantial. In such cases the maximum value of $\gamma$ should be set low, so that $n$ can be safely set to low values.

### 3.7.1 Mean and variance

Due to the lack of a closed form expression for the tails, we cannot provide a similar result to Proposition 1. However, we can show that the expectation converges to $r(\mu)$ as $\gamma \to 0$ and that the variance converges to 0 (with the same exceptions as Dalap). Let $c$ and $f$ be defined as they were for Dalap.

**Proposition 4.** *If $\lfloor \mu \rfloor, \lceil \mu \rceil \in S$ then $\mathcal{E}_{n \sim p(n|n \in S, \mu, \gamma)} n$ goes to $r(\mu)$ as $\gamma \to 0$ unless $c = f$ in which case it goes to $\mu$.*

*Proof.* If $\mu$ is an integer the result follows by inspection in the same manner as it did for Dalap.

If $\mu$ is non-integer, we first write $\mathcal{E}n = \sum_{n \in S} p(n) n$. As with Dalap, we would like to show that only the direct neighbors of $\mu$ get nonzero probability as $\gamma \to 0$.

Let $l$ and $r$ be the number of steps the support extends to the left and right of the neighbors of $\mu$ (with $\infty$ a possibility).

$$\lim_{\gamma \to 0} \mathcal{E}n = \lim \sum_n \frac{1}{z} \gamma^{(n - \mu)^2} n = \frac{\lim \sum_n n \gamma^{(n - \mu)^2}}{\lim \sum_n \gamma^{(n - \mu)^2}} = \frac{\sum_n n \lim \gamma^{(n - \mu)^2}}{\sum_n \lim \gamma^{(n - \mu)^2}}$$

$$= \frac{\sum_{i=0}^l (i + \lfloor \mu \rfloor) \lim \gamma^{(i + f)^2} + \sum_{i=0}^r (i + \lceil \mu \rceil) \lim \gamma^{(i + c)^2}}{\sum_{i=0}^l \lim \gamma^{(i + f)^2} + \sum_{i=0}^r \lim \gamma^{(i + c)^2}}.$$

---

[7]To stabilize the implementation, we actually compute $\log \tilde{z}$ by computing the log-values of the terms of the sum and summing with the log-sum-exp trick. Out-of-bounds values are set to $-\infty$. See Appendix A.5 for further details.

Now, for any of the limits in green, we can write $\gamma^{(i+c)^2} = \gamma^{ii}\gamma^{2ci}\gamma^{cc}$, which gives us for the limit

$$0 \leq \lim \gamma^{ii}\gamma^{2ci}\gamma^{cc} \leq \lim \gamma\gamma^{2c}\gamma^{cc} = 0$$

for any $i \geq 1$, and likewise for the ones containing $f$.

Eliminating these terms, we are left with

$$\frac{\lfloor\mu\rfloor \lim \gamma^0\gamma^{2f0}\gamma^{ff} + \lceil\mu\rceil \lim \gamma^0\gamma^{2c0}\gamma^{cc}}{\lim \gamma^0\gamma^{2f0}\gamma^{ff} + \lim \gamma^0\gamma^{2c0}\gamma^{cc}} = \lim \frac{\gamma^{ff}}{\gamma^{ff} + \gamma^{cc}}\lfloor\mu\rfloor + \frac{\gamma^{cc}}{\gamma^{ff} + \gamma^{cc}}\lceil\mu\rceil.$$

Rewriting the probabilities as $\frac{\gamma^{f^2}}{\gamma^{f^2}+\gamma^{c^2}} = \frac{1}{1+\gamma^{c^2-f^2}}$ we see that the limit for $\gamma \to 0$ is the same as in (3), if $c > f$ it goes to 0, if $c = f$ to $\frac{1}{2}$ and if $c < f$ to 1, resulting in the same expectation in the limit as we found for Dalap. $\qquad\square$

**Proposition 5.** *The variance of Danorm goes to* 0 *as* $\gamma \to 0$, *unless* $c = f$ *in which case it goes to* 1/4.

*Proof.* We follow the proof for Proposition 3. In the sum, we use the logic from Proposition 4 to isolate only the terms for $\lfloor\mu\rfloor$ and $\lceil\mu\rceil$, resulting in

$$\lim_{\gamma\to 0} \mathcal{E}_n n^2 - (\mathcal{E}_n n)^2 = \frac{\gamma^{f^2}}{\gamma^{f^2} + \gamma^{c^2}}\lfloor\mu\rfloor^2 + \frac{\gamma^{c^2}}{\gamma^{f^2} + \gamma^{c^2}}\lceil\mu\rceil^2 - r(\mu)^2$$

As established, the limits of these probabilities are the same as their equivalents in the proof of Proposition 3, so the same result follows. $\qquad\square$

### 3.8 Bitwise

The *Bitwise* distribution is based on the common practice in neural network modeling of representing integers by their binary representation. While this approach has been used for a long time—for integers or other large discrete output spaces Dietterich & Bakiri (1995); Sejnowski & Rosenberg (1986)—we believe this is the first *interpretation* of this approach as probability distribution on the integers.

First, let $\mathbb{B}^k$ represent all bitstrings $\boldsymbol{x}$ of length $k$. We then define the function $d : \mathbb{B}^k \to \mathbb{Z}$ as

$$d(\boldsymbol{x}) = (2x_1 - 1)\left(x_2 2^0 + x_3 2^1 + \ldots + x_k 2^{k-2}\right).$$

That is, $d(\boldsymbol{x}) = n$ decodes a bitstring to an integer using signed-magnitude representation. We use $e(n) = \boldsymbol{x}$ for the inverse operation of *encoding* an integer into a bitstring.

Next, we define a parameter vector $\boldsymbol{\pi}$ with $k$ real-valued elements $\pi_i \in [0, 1]$. We define our distribution on $\mathbb{Z}$ first from a sampling perspective. For a distribution with parameters $\boldsymbol{\pi}$ we sample from the corresponding Bitwise distribution, by treating each $\pi_i$ as a Bernoulli distribution over the values 0 and 1. Sampling from each independently yields a bitstring $\boldsymbol{x}$, which we decode as $d(\boldsymbol{x}) = n$ to obtain the integer $n$.

The corresponding probability mass function $q(n \mid \boldsymbol{\pi})$, by independence is defined simply as

$$q(n \mid \boldsymbol{\pi}) = \prod_{i=1}^{k} \mathrm{Bern}(x_i \mid \pi_i) \qquad\qquad \text{with } x = e(n)$$
$$= \prod_i \pi_i^{x_i}(1 - \pi_i)^{1-x_i} \ .$$

We must, however, make one exception for $n = 0$, which has two possible encodings under $e$—one for $-0$ and one for $+0$. To keep our notation simple, we assume that $q$ is supported by the set of integers with a signed zero. We can then define a distribution on $\mathbb{Z}$ as

$$p(n \mid \boldsymbol{\pi}) = \begin{cases} q(+0) + q(-0) & \text{if } n = 0 \\ q(n) & \text{otherwise.} \end{cases}$$

In practice, we always encode 0 as $+0$. This means that a small amount of probability mass is always expended on unseen events.[8]

### 3.8.1 Nonnegative support

Bitwise always has a bounded support of $[-2^{k-1} + 1, 2^{k-1} - 1]$. For a version only supported on $[0, 1, 2, \ldots)$, we remove the sign from the representation, using

$$d(\boldsymbol{x}) = x_1 2^0 + x_2 2^1 + \ldots + x_k 2^{k-1} .$$

resulting in a support of $[0, 2^k - 1]$. We refer to the first version of Bitwise as *unbounded* and the second as *nonnegative*.

While a support on all of $\mathbb{Z}$ or $\mathbb{N}$ is impossible, the size of the support grows exponentially in $k$, which means that we can usually set $k$ large enough to deal with any integers $n$ we are likely to encounter. We therefore consider bitwise unbounded for practical purposes.[9]

For a two-way bounded version of Bitwise, we can simply choose the smallest interval that takes the form $[-2^{k-1} + 1, 2^{k-1} - 1]$ or $[0, 2^k - 1]$ which covers the required interval. Unlike Dalap and Danorm, Bitwise does not allow the support to perfectly coincide with all possible bounds.

### 3.8.2 Mean and variance

As with Dalap, we show a brief analysis of the mean and variance, to show that the mean is easily computable from the parameters and that the variance can be made arbitrarily small. Let round($\boldsymbol{x}$) be the function that rounds each element of the vector $\boldsymbol{x}$ to its nearest integer.

Let $\pi_i^{\bullet} = 1 - \pi_i$ and $\pi_i^{\boldsymbol{x}} = \begin{cases} \pi_i & \text{if } x_i = 1 \\ \pi_i^{\bullet} & \text{if } x_i = 0 \end{cases}$ and let $\boldsymbol{\pi}^{\boldsymbol{x}} = \pi_1^{\boldsymbol{x}} \times \ldots \times \pi_k^{\boldsymbol{x}}$. Let $\boldsymbol{\pi}_{a:b}$ represent the subvector of $\boldsymbol{\pi}$ from $a$ to $b$ inclusive. If $a$ or $b$ is omitted the subvector is taken from the start or end of $\boldsymbol{\pi}$ respectively.

**Proposition 6.** *The mean of a nonnegative Bitwise distribution with parameters $\boldsymbol{\pi}$ is*

$$\pi_1 2^0 + \ldots + \pi_k 2^{k-1}.$$

---

[8]This could be avoided by using a two's complement encoding instead. However, for the upstream model we expect that sign-magnitude representation offers a better separation of concerns, with the magnitude bits representing a magnitude in the same way for both negative and positive integers.

[9]Note that in settings where this is not the case, we would already be unable to represent our data in computer memory using standard integer representations.

*Proof.* We proceed by induction. For the base case where $k = 1$, the expected value is $\pi_1 1 + \pi_1^{\bullet} 0 = \pi_1 2^0$. For the inductive case:

$$
\begin{aligned}
\mathcal{E}_{n \sim p(x|\boldsymbol{\pi})} n &= \sum_{n \in [0,\infty)} p(n \mid \boldsymbol{\pi}) n = \sum_{\boldsymbol{x} \in \mathbb{B}^k} \boldsymbol{\pi}^{\boldsymbol{x}} d(\boldsymbol{x}) \\
&= \sum_{\boldsymbol{x}} \boldsymbol{\pi}^{\boldsymbol{x}} (x_1 2^0 + \ldots + x_k 2^{k-1}) = \sum_{\boldsymbol{x},i} \boldsymbol{\pi}^{\boldsymbol{x}} x_i 2^{i-1} \\
&= \sum_{i, \boldsymbol{x} \in \mathbb{B}^k_{k=1}} \boldsymbol{\pi}^{\boldsymbol{x}} x_i 2^{i-1} + \sum_{i, \boldsymbol{x} \in \mathbb{B}^k_{k=0}} \boldsymbol{\pi}^{\boldsymbol{x}} x_i 2^{i-1} && \color{blue}{\mathbb{B}^k_{i=v} = \{\boldsymbol{x} \in \mathbb{B}^k : \boldsymbol{x}_i = v\}} \\
&= \pi_k \cdot 1 \cdot 2^{k-1} + \pi_k \sum_{i \in [1,k-1], \boldsymbol{x} \in \mathbb{B}^{k-1}} \boldsymbol{\pi}^{\boldsymbol{x}}_{:k-1} x_i 2^{i-1} \\
&\quad + \pi_k^{\bullet} \cdot 0 \cdot 2^{k-1} + \pi_k^{\bullet} \sum_{i \in [1,k-1], \boldsymbol{x} \in \mathbb{B}^{k-1}} \boldsymbol{\pi}^{\boldsymbol{x}}_{:k-1} x_i 2^{i-1} \\
&= \pi_k 2^{k-1} + (\pi_k + \pi_k^{\bullet}) \sum_{i \in [1,k-1], \boldsymbol{x} \in \mathbb{B}^{k-1}} \boldsymbol{\pi}^{\boldsymbol{x}}_{:k-1} x_i 2^{i-1} && \color{red}{\text{note } \pi_k + \pi_k^{\bullet} = 1} \\
&= \pi_k 2^{k-1} + \mathcal{E}_{n \sim p(x|\boldsymbol{\pi}_{:k-1})} n.
\end{aligned}
$$

By the inductive hypothesis, $\mathcal{E}_{n \sim p(x|\boldsymbol{\pi}_{:k-1})} n$ is equal to the required remainder of the sum. $\qquad\square$

Note two useful edge cases. In the case that all $\pi_i = \frac{1}{2}$ we obtain a uniform distribution over $\mathbb{B}^k$ and thus over $[0, 2^k - 1]$. In the case that $\boldsymbol{\pi} = \boldsymbol{x}$ we obtain a deterministic distribution, where $p(d(\boldsymbol{x})) = 1$.

We next analyze the unbounded case. For ease of notation, we indicate the parameter for the sign bit with $\pi_{\text{pos}}$ and include only the remaining parameters in the vector $\boldsymbol{\pi}$ (and likewise for $\boldsymbol{x}$). Let $k$ be the length of $\boldsymbol{\pi}$ thus defined.

**Proposition 7.** *For an unbounded Bitwise distribution with parameters $\pi_{pos}, \boldsymbol{\pi}$, the expected value is*

$$
(\pi_{pos} - \pi_{pos}^{\bullet})(\pi_1 2^0 + \ldots \pi_k 2^{k-1}).
$$

*Proof.*

$$
\begin{aligned}
\mathcal{E}_{n \sim p(x|\boldsymbol{\pi})} n &= \sum_{n \in (-\infty,\infty)} p(n \mid \boldsymbol{\pi}) n = \sum_{\boldsymbol{x} \in \mathbb{B}^k} \pi_{\text{pos}}^{\boldsymbol{x}} \boldsymbol{\pi}^{\boldsymbol{x}} d(\boldsymbol{x}) \\
&= \sum_{\boldsymbol{x}} \pi_{\text{pos}}^{\boldsymbol{x}} \boldsymbol{\pi}^{\boldsymbol{x}} (2x_{\text{pos}} - 1)(x_1 2^0 + \ldots + x_k 2^{k-1}) \\
&= \sum_{\boldsymbol{x}} \pi_{\text{pos}}^{\boldsymbol{x}} \boldsymbol{\pi}^{\boldsymbol{x}} (2x_{\text{pos}} - 1) \sum_i x_i 2^{i-1} \\
&= \pi_{\text{pos}} \cdot 1 \cdot \sum_{i, \boldsymbol{x} \in \mathbb{B}^k} \boldsymbol{\pi}^{\boldsymbol{x}} x_i 2^{i-1} + \pi_{\text{pos}}^{\bullet} \cdot -1 \cdot \sum_{i, \boldsymbol{x} \in \mathbb{B}^k} \boldsymbol{\pi}^{\boldsymbol{x}} x_i 2^{i-1} \\
&= (\pi_{\text{pos}} - \pi_{\text{pos}}^{\bullet}) \sum_{i, \boldsymbol{x}} \boldsymbol{\pi}^{\boldsymbol{x}} x_i 2^{i-1} = (\pi_{\text{pos}} - \pi_{\text{pos}}^{\bullet}) \mathcal{E}_{n \sim p(x|\boldsymbol{\pi}, n \geq 0)} n
\end{aligned}
$$

Substituting the result from Proposition 6 completes the proof. $\qquad\square$

Next, we show that the variance has a predictable form, which can be made arbitrarily small around an arbitrary value in the support.

**Proposition 8.** *The variance of both the unbounded and the non-negative Bitwise distribution goes to $0$ as $\boldsymbol{\pi} \to \bar{\boldsymbol{x}}$ for any $\bar{\boldsymbol{x}}$.*

*Proof.* We write the variance as $\mathcal{E}(n - \hat{n})^2$ with $\hat{n}$ the mean. This gives us for the **nonnegative** distribution

$$
\begin{aligned}
\mathcal{E}(n - \hat{n})^2 &= \sum_{\boldsymbol{x} \in \mathbb{B}^k} \boldsymbol{\pi}^{\boldsymbol{x}} (d(x) - \hat{n})^2 \\
&= \sum_{\boldsymbol{x} \in \mathbb{B}^k} \boldsymbol{\pi}^{\boldsymbol{x}} (x_1 2^0 + \ldots + x_k 2^{k-1} - \pi_1 2^0 - \ldots - \pi_k 2^{k-1})^2 \\
&= \sum_{\boldsymbol{x} \in \mathbb{B}^k} \boldsymbol{\pi}^{\boldsymbol{x}} \left( (x_1 - \pi_1) 2^0 + \ldots + (x_k - \pi_k) 2^{k-1} \right)^2 .
\end{aligned}
$$

As $\boldsymbol{\pi} \to \bar{\boldsymbol{x}}$ the value $\boldsymbol{\pi}^{\boldsymbol{x}}$ goes to 0 for any $\boldsymbol{x} \neq \bar{\boldsymbol{x}}$ to 1 for $\boldsymbol{x} = \bar{\boldsymbol{x}}$, causing all terms but the one corresponding to $\bar{\boldsymbol{x}}$ to vanish. In that remaining term, the factors $x_i - \pi_i$ go to zero, since $\pi_i \to x_i$, causing this final term also to vanish, resulting in a variance of 0.

For the **unbounded** distribution, we have

$$
\begin{aligned}
\mathcal{E}(n - \hat{n})^2 &= \sum_{x_{\text{pos}} \in \mathbb{B}, \boldsymbol{x} \in \mathbb{B}^k} \pi_{\text{pos}} \boldsymbol{\pi}^{\boldsymbol{x}} (d(x_{\text{pos}} \boldsymbol{x}) - \hat{n})^2 \\
&= \sum_{x_{\text{pos}}, \boldsymbol{x}} \pi_{\text{pos}} \boldsymbol{\pi}^{\boldsymbol{x}} ((2x_{\text{pos}} - 1)(x_1 2^0 + \ldots + x_k 2^{k-1}) - (\pi_{\text{pos}} - \pi_{\text{pos}}^{\bullet})(\pi_1 2^0 + \ldots + \pi_k 2^{k-1}))^2 \\
&= \pi_{\text{pos}} \sum_{\boldsymbol{x}} \boldsymbol{\pi}^{\boldsymbol{x}} (1 \cdot (x_1 2^0 + \ldots + x_k 2^{k-1}) - (\pi_{\text{pos}} - \pi_{\text{pos}}^{\bullet})(\pi_1 2^0 + \ldots + \pi_k 2^{k-1}))^2 \\
&\quad + \pi_{\text{pos}}^{\bullet} \sum_{\boldsymbol{x}} \boldsymbol{\pi}^{\boldsymbol{x}} (-1 \cdot (x_1 2^0 + \ldots + x_k 2^{k-1}) - (\pi_{\text{pos}} - \pi_{\text{pos}}^{\bullet})(\pi_1 2^0 + \ldots + \pi_k 2^{k-1}))^2 \\
&= \pi_{\text{pos}} \sum_{\boldsymbol{x}} \boldsymbol{\pi}^{\boldsymbol{x}} ((x_1 - (\pi_{\text{pos}} - \pi_{\text{pos}}^{\bullet})\pi_1) 2^0 + \ldots + (x_k - (\pi_{\text{pos}} - \pi_{\text{pos}}^{\bullet})\pi_k) 2^{k-1})^2 \\
&\quad + \pi_{\text{pos}}^{\bullet} \sum_{\boldsymbol{x}} \boldsymbol{\pi}^{\boldsymbol{x}} ((-x_1 - (\pi_{\text{pos}} - \pi_{\text{pos}}^{\bullet})\pi_1) 2^0 + \ldots + (-x_k - (\pi_{\text{pos}} - \pi_{\text{pos}}^{\bullet})\pi_k) 2^{k-1})^2
\end{aligned}
$$

As $\boldsymbol{\pi} \to \boldsymbol{x}$, we first check whether $\boldsymbol{x}_{\text{pos}} = 1$. If so, then the second term vanishes. In the first term, each factor $x_i - (\pi_{\text{pos}} - \pi_{\text{pos}}^{\bullet})\pi_i$ reduces to $x_i - \pi_i \to 0$, making the total sum zero. If $\boldsymbol{x}_{\text{pos}} = 0$, the first term vanishes, and in the second term, each factor $-x_i - (\pi_{\text{pos}} - \pi_{\text{pos}}^{\bullet})\pi_i$ reduces to $-x_i + \pi_i \to 0$, and the total sum again goes to 0. $\qquad\square$

## 4 Experiments

We conduct a series of experiments to evaluate the effectiveness of the different distribution functions in modeling integer-valued labels. We use tabular data, sequential time-series data, and image data to provide a comprehensive evaluation. The choice of datasets, including Bicycles, Upvotes, Migration, MAESTRO, MNIST, FashionMNIST and CIFAR10, ensures that the models are tested on the bounded, unbounded, and one-way bounded methods.

In all experiments, we aim to answer two main questions:

1. Assuming a discrete distribution on the labels is required, which of the six candidates performs best in terms of negative log-loss?

2. Assuming a continuous relaxation is also an option, do any of the candidates outperform such a relaxation in terms of RMSE?

For the tabular regression and MAESTRO experiments, each experiment is executed with a sweep over learning rates: $3.4 \cdot 10^{-3}$, $1 \cdot 10^{-3}$, $3.4 \cdot 10^{-4}$, $1 \cdot 10^{-4}$, $3.4 \cdot 10^{-5}$, and $1 \cdot 10^{-5}$. For these experiments, we also

test mixture models with $k \in \{1, 2, 4, 8\}$, where $k = 1$ is the standard unmixed distribution without an extra mixture-weight output. For the image modeling experiments, due to computational cost, we evaluate each distribution using a single component and a mixture model with $K = 10$, as described in the PixelCNN++ experiments below. Across experiments, hyperparameters are selected based on the log loss on the validation set, and results are reported on the test set. For the tabular regression and MAESTRO experiments, results are reported as the mean $\pm$ standard error (SE) over 10 random seeds. The dataset is kept fixed between runs; that is, we do not re-split or resample. For the image modeling experiments, the substantially higher computational cost of training PixelCNN models, each requiring multiple days on a single GPU, makes multiple training runs impractical; we therefore report only the results of a single run.

In addition to the distribution functions introduced in this work, we include several baselines. The **Poisson** distribution serves as a natural baseline for one-way bounded (non-negative) count data. Since the Poisson is defined only on $\mathbb{Z}_{\geq 0}$, it cannot be applied to unbounded targets such as net migration, nor to the bounded pixel range $[0, 255]$ in the image experiments; Poisson results are therefore reported only for the Bicycles, Upvotes and MAESTRO datasets. A **Categorical** distribution is included for settings with known, finite support. The categorical treats each integer as an independent class, ignoring the ordinal structure of the data, and cannot generalize to values outside the observed range, making it unsuitable for unbounded targets where unseen values may appear at test time. We report categorical results only in the non-mixture image experiments: the number of parameters scales linearly with both the support size and the number of mixture components $K$, leading to prohibitive cost without commensurate gains for $K > 1$. The **discretized logistic** (Dlogistic) from PixelCNN++ (Salimans et al., 2017) serves as the primary comparison for the image experiments. Dlogistic requires a predefined bounded support for discretization, which is naturally available for pixel values ($[0, 255]$) but impractical for the tabular targets with no fixed bounds or with ranges spanning $\pm 10^6$.

## 4.1 Tabular regression

We use the following datasets

The **Bicycles** dataset Fanaee-T (2013) focuses on predicting the hourly number of bicycles parked in a designated space within the Capital Bike Sharing System of Washington D.C. initially comprising 12 features, one-hot encoding of nominal features expands the dataset to 58 features. The dataset is divided into a training set (10,948 samples, 63%), a test set (5,214 samples, 30%), and a validation set (1,217 samples, 7%). The target variable exhibits moderate overdispersion with a dispersion index (DI $= \sigma^2/\mu$) of 173.66 and a skewness of 1.28, indicating a right-skewed distribution with variance substantially exceeding the mean.

In the **Upvotes** dataset Naseer (2020), the task is to predict the number of upvotes a social media post will receive. It employs a similar train/test/validation split as the Bicycles dataset, with 207,927 training samples, 99,014 test samples, and 23,104 validation samples. The primary goal is to predict upvotes based on the post's tag, number of answers, and the poster's reputation. One-hot encoding of the tag type (the only nominal feature) and standard scaling of other features results in a total of 12 features. This dataset displays extreme overdispersion (DI $= 38,238.31$) with heavy tails (excess kurtosis $= 8,919.66$) and severe right-skewness (74.25), characteristic of viral social media dynamics where most posts receive few upvotes while a small fraction achieve disproportionately high engagement.

The **Migration** dataset is a combination of various datasets of the World Bank Group World Bank (2024). This dataset has as a target the net migration net (2022) with different features from the datasets: agricultural development agr (2018), environmental indicators env (2024), climate change indicators cli (2024) and the financial sector fin (2024). This leaves 16,565 instances after combining these different datasets, this merged dataset still includes 8,620 instances which are dropped leaving with a complete dataset of 7,945 instances. Furthermore, the country names are factorized and all features except the country name, year and net migration are Standard Scaled. This gives 4,879 train samples, 2,066 test samples and 1,000 validation samples. Unlike the count-based targets of the previous datasets, net migration includes both positive and negative values with a mean close to zero ($\mu = 1,601.57$) and exhibits extreme variance ($\sigma^2 = 2.96 \times 10^{10}$), resulting in heavy tails (excess kurtosis $= 50.21$) driven by outlier countries with large migration flows.

All three datasets exhibit significant overdispersion, making them suitable benchmarks for evaluating distribution functions capable of capturing varying degrees of dispersion. A detailed summary of the target distributions, including histograms and dispersion characteristics, is provided in Appendix A.1.

For all tabular datasets, we use a multi-layer perceptron (MLP) with ReLU activation functions. The first layer has an input size of $n$, the second layer a hidden size of 128 and the output layer a size of 1 in the case of the squared error, 32 in the case of Bitwise and 2 for the rest of the distributions. The mapping from raw network outputs to valid distribution parameters is detailed in Appendix A.4.

## 4.2 MAESTRO

The **MAESTRO** (V3) dataset Hawthorne et al. (2019) consists of 200 hours of MIDI recordings of professional piano players playing challenging pieces of classical music captured on a digital Piano.

MAESTRO is a curated dataset of classical piano music featuring various levels of expressiveness, widely considered the gold standard for evaluating generative music models. For this study, the dataset yields 13,914,411 training, 4,423,513 test, and 1,488,572 validation samples. The number of sequences per file was calculated by subtracting the context window length (128 tokens) from the file's total token count.

We stop short of building a full generative model for MIDI music, leaving that to future work, and focus only on the task of predicting at any moment in time, *the time in ticks*[10] *for which the next note will be held*, given the sequence of $n$ MIDI events observed just before.

The MIDI data is encoded in six different events: Event (see table 1), Data1 (Pitch), Data2 (Velocity), Instrument, Channel, and Tick. The first five are embedded with an embedding size of 128. The Tick events are encoded using a linear layer where the tick integer, transformed to a binary string (32 bits) is concatenated with the log value of the tick. In table 1 an overview of the value ranges can be found.

Table 1: MIDI representation

| Event | Data 1 | Data 2 | Channel | Instrument | MIDI Tick |
|---|---|---|---|---|---|
| Note on | 0-127 | 0-127 | 0-15 | 0-127 | 0-$\infty$ |
| Note off | 0-127 | 0-127 | 0-15 | 0-127 | 0-$\infty$ |
| Polytouch | 0-127 | 0-127 | 0-15 | 0-127 | 0-$\infty$ |
| Pitch Bend | 0-127 | 0-127 | 0-15 | 0-127 | 0-$\infty$ |
| Channel Pressure | 0-127 | 0-127 | 0-15 | 0-127 | 0-$\infty$ |

To make the predictions, we use an LSTM. The model consists of 2 LSTM layers, each with a hidden dimension of 128. The context length is 128 and the model is trained for 100 epochs. Depending on the distribution target (similarly to the MLP) 2 outputs are used for the distributions, 1 output is used for squared error and 16 for Bitwise (since the highest number in the dataset does not exceed 16 bits there is no need to go over 16 bits in the output). During training, the per-bit binary cross-entropy loss for Bitwise is weighted by $2^i$ for bit position i, giving higher-order bits proportionally more influence on the gradient. During evaluation, all bits are weighted equally. This weighting reflects the fact that errors in higher-order bits have a larger impact on the decoded integer value. A positional weighting scheme is applied to the bitwise training loss (Appendix A.6).

The model is evaluated every 5000 training samples with a subset of 500 validation samples. After training, we evaluate on the full test data.

During training, the loss is calculated over the entire output sequence and the mean is taken over the batch and sequence. During validation and testing the loss is calculated over only the next token in the sequence which are then summed and averaged over the number of instances.

---

[10]The *tick* is a unit of time specific to MIDI music. The length of a tick can be different between MIDI files, but a common choice is 5ms per tick.

### 4.3 PixelCNN++

Finally, we test our distributions in the task of image generation. As noted in the introduction, this requires a prediction in the range $[0, 255]$ for every color channel of every pixel, making it a good setting to test the two-way bounded distributions.

The current state of the art in this domain is almost exclusively built on diffusion models. However, diffusion is not the most natural place to test our integer models. In Ho et al. (2020), for example, a separate decoder module is required in the last generation step to map the continuous representation of the image to an integer representation. Even then, the authors note

> Still, while our lossless codelengths are better than the large estimates reported for energy based models and score matching using annealed importance sampling [...], they are not competitive with other types of likelihood-based generative models [...]. Since our samples are nonetheless of high quality, we conclude that diffusion models have an inductive bias that makes them excellent lossy compressors.

The discrete decoder was dispensed with in most follow-up work like Dhariwal & Nichol (2021) and diffusion models in general produce outputs in a continuous space, with evaluation performed solely on metrics like IS and FID.

Since we are interested in using compression (that is, the negative log loss) as our main metric of evaluation—with perceptual quality a secondary consideration—we will focus on likelihood-based generative models, specifically pixel-wise auto-regressive models. So far, these have often used categorical distributions van den Oord et al. (2016) which ignore the ordinal aspect of the data, or various continuous relaxations Loaiza-Ganem & Cunningham (2019); Ramesh et al. (2021).

A notable advancement in this domain is PixelCNN++ Salimans et al. (2017), which uses discretized logistic mixtures to model pixel values. While this approach improves image quality and lowers Bits Per Dimension (BPD) compared to the original PixelCNN, it relies on an underlying continuous density function that is subsequently discretized. In this work, we adopt the architectural backbone of PixelCNN++, specifically the use of padding and shifting over masked kernels, as well as the use of coefficients to maintain RGB ordinality. With this architecture we adopt several distribution functions to compare against the discretized logistic mixture model.

Note that the aim here is not to achieve state-of-the-art compression loss, or image generation. We simply want to compare the different output distributions on a relatively level playing field, by fixing the upstream architecture. For this purpose, the PixelCNN++ model provides a good balance between performance and required resources.

In Salimans et al. (2017), the authors bridge the gap between continuous predictions and discrete pixel values by integrating the logistic distribution over bins, effectively rounding the continuous values, as in the discretized normal and Laplace distributions above. As we saw in the tabular data, discrete analogues often work better than discretization. We therefore hypothesize that Dalap improves stability during both training and generation by eliminating the need for discretization or rounding.

We exclude Dweib and Danorm from the PixelCNN experiments due to practical limitations. Dweib exhibited numerical instability during preliminary experiments, particularly when modeling the bounded $[0, 255]$ range required for pixel values, leading to frequent convergence failures. Danorm, while theoretically sound, requires computing a numerical approximation of the partition function by summing over a range of integers (as described in Section 3.7). For image generation, this computation must be performed for every pixel in every image in the batch, requiring large amounts of memory. In our experiments, Danorm required over 90GB of VRAM, exceeding our available hardware capacity, making it impractical for image modeling tasks. We also exclude the categorical distribution from the mixture image experiments: the number of parameters scales linearly with both the support size and $K$, resulting in prohibitive computational cost without performance gains over $K = 1$.

To validate this, we compare our functions against the original PixelCNN++ results. We evaluate performance using two metrics: Bits Per Dimension (BPD) to measure compression and log-likelihood quality, and the Fréchet Inception Distance (FID) to assess the perceptual quality of the generated images and their similarity to the input distribution across the domains of reconstruction, seeded sampling and random sampling. Furthermore, we investigate whether our method can achieve superior or equal generative performance (lower FID) and likelihood scores (lower BPD) with fewer parameters, thereby demonstrating the efficiency of modeling integer labels with truly discrete distributions.

We evaluate the generative capabilities of our models in three distinct ways:

1. *Unconditional Generation*: Sampling is performed starting from an empty canvas (initialized to a value of -1).

2. *Image Completion*: The model is seeded with the first 8 rows of a ground-truth image from the dataset, and generates the remaining pixels.

3. *Reconstruction*: The model regenerates images based on full input representations.

We use three common low-resolution datasets: MNIST LeCun et al. (1998) which contains handwritten digits, FashionMNIST Xiao et al. (2017) which contains images of items of clothing and CIFAR10 Krizhevsky & Hinton (2009), which contains color images of ten diverse classes of objects.

Training follows the hyperparameters described in Salimans et al. (2017). We use Exponential Moving Average (EMA), and a learning rate decay of 0.99995, with an initial learning rate of $10^{-3}$ for CIFAR10 and $10^{-4}$ for FashionMNIST and MNIST. We evaluate each distribution function using both a single component and a mixture of 10 components. All models are trained for 300 epochs across all datasets.

## 5 Results

### 5.1 Count data regression

We evaluate several distribution functions for integer-valued regression on four datasets: Bicycles, Upvotes, Migration, and Maestro. Performance is reported in Tables 2 and 3 using two metrics: Bits (negative log-likelihood in base 2; lower is better) and RMSE (accuracy of the predicted mean; lower is better).

Dalap achieves the lowest bits on Bicycles ($6.78 \pm 0.02$) and Upvotes ($6.74 \pm 0.01$) in the non-mixture setting. On MAESTRO, Dalap again achieves the lowest bits ($5.00 \pm 0.00$), reinforcing its strong performance across datasets with varying dispersion levels. Poisson, despite being a natural baseline for non-negative count data, performs poorly on all applicable datasets (11.4, 56.6, and 16.4 bits on Bicycles, Upvotes, and MAESTRO respectively), confirming that its single-parameter form cannot capture the variance structure present in these targets. Notably, Poisson still achieves competitive RMSE on Bicycles and MAESTRO, even obtaining the best RMSE among all distributions in the mixture setting on Bicycles ($59.0 \pm 0.9$), suggesting that its mean prediction can be reasonable even when the distributional fit is poor. When examining the RMSE scores, Dalap benefits considerably from multiple mixture components, achieving competitive RMSE performance while maintaining strong bits scores.

The results show variability in the performance across datasets, suggesting that practitioners should test different output distributions for their data. However, if the negative log likelihood is the key metric, then Dalap is likely a good candidate in most settings.

On the Migration dataset, Bitwise achieves the best bits in both the non-mixture setting ($22.9 \pm 1.0$) and the mixture setting ($18.0 \pm 0.0$). Dalap without mixture components shows sensitivity to initialization on this dataset: 8 of 10 seeds converge to low bits values, while 2 seeds diverge to substantially higher losses, resulting in a large standard error ($2.9 \times 10^5 \pm 1.9 \times 10^5$). This reflects the challenge of modeling data with extreme dispersion (DI $> 10^7$) and heavy tails (excess kurtosis = 50.21) using a single component. This instability is effectively resolved by the mixture model: with $K = 8$ components, Dalap achieves $20.4 \pm 1.0$ bits with stable convergence across all seeds, placing it close to Bitwise ($18.0 \pm 0.0$). The strong performance

of Bitwise on Migration, combined with its weaker results on the other datasets, suggests that the best distribution depends on the characteristics of the target data, with Migration's extreme variance favoring these alternatives over Dalap.

Furthermore, our other distribution function, Bitwise, performs well in terms of bits on highly dispersed data such as Migration (as discussed above). However, it exhibits significantly increased RMSE scores compared to other distribution functions across all datasets. It is likely that some probability assigned to higher-order bits causes large outliers to be assigned small but non-zero probability, to which the RMSE is sensitive. Whether this is a problem depends largely on the use case for which the model is deployed.

The Dlaplace, Dnormal, and Danorm functions show related behavior across some datasets, which is expected given their structural similarities in how the distributions are formulated. All three achieve reasonable performance in some settings, but none of them consistently dominates across datasets. Danorm is conceptually close to squared-error regression, but in these experiments its rapidly decaying tails often hurt negative log-likelihood and its RMSE does not consistently improve over the other alternatives.

Finally, Dweib underperforms across multiple metrics, demonstrating its practical limitations. It is restricted to non-negative support, precluding its use on the Migration dataset, and exhibits numerical instability during training, leading to poor performance on the Upvotes dataset ($130.3 \pm 13.6$ bits).

When comparing against the squared error baseline, which uses a continuous relaxation, we observe that the continuous approach achieves the best RMSE in three out of four datasets in the non-mixture setting. In the mixture setting, however, discrete mixture models outperform squared error on some datasets, showing that the RMSE advantage of continuous relaxation is not uniform. This suggests that when RMSE is the primary concern and a discrete distribution is not strictly required, continuous relaxations remain strong baselines, but they are not guaranteed to provide the best performance once mixture models are considered. However, when a proper discrete distribution is needed, for example to compute valid log-probabilities or to combine losses from discrete and continuous predictions, Dalap likely offers the best balance of bits and RMSE performance.

Table 2: Test set results for bicycles, upvotes, migration and maestro. Seeded results reported as mean $\pm$ SE over 10 seeds; best in bold.

| | Bicycles | | Upvotes | | Migration | | MAESTRO | |
|---|---|---|---|---|---|---|---|---|
| | Bits | RMSE | Bits | RMSE | Bits | RMSE | Bits | RMSE |
| Dalap | $\mathbf{6.78 \pm 0.02}$ | $128 \pm 1$ | $\mathbf{6.74 \pm 0.01}$ | $\mathbf{126 \pm 3}$ | $2.9e5 \pm 1.9e5$ | $9.0e4 \pm 2.6e4$ | $\mathbf{5.00 \pm 0.00}$ | $53.9 \pm 0.0$ |
| Bitwise | $8.13 \pm 0.04$ | $1.5e4 \pm 2.2e3$ | $6.86 \pm 0.00$ | $3400 \pm 654$ | $\mathbf{22.9 \pm 1.0}$ | $2.1e5 \pm 1.6e4$ | $5.17 \pm 0.01$ | $45.1 \pm 1.9$ |
| Dlaplace | $7.04 \pm 0.04$ | $64.1 \pm 0.6$ | $7.18 \pm 0.01$ | $370 \pm 9$ | $48.1 \pm 7.5$ | $1.1e5 \pm 2.4e4$ | $5.42 \pm 0.00$ | $47.8 \pm 0.1$ |
| Dnormal | $6.87 \pm 0.01$ | $65.9 \pm 0.3$ | $7.09 \pm 0.00$ | $513 \pm 18$ | $23.2 \pm 0.0$ | $1.7e5 \pm 2.9e3$ | $5.58 \pm 0.00$ | $51.1 \pm 0.0$ |
| Dweib | $9.15 \pm 0.85$ | $112 \pm 6$ | $130.3 \pm 13.6$ | $3026 \pm 177$ | — | — | $7.88 \pm 1.18$ | $74.0 \pm 3.2$ |
| Danorm | $7.40 \pm 0.06$ | $142 \pm 8$ | $8.42 \pm 0.13$ | $1584 \pm 71$ | $4.2e3 \pm 5.3e2$ | $1.5e5 \pm 1.2e4$ | $5.27 \pm 0.01$ | $49.3 \pm 0.3$ |
| Poisson | $11.4 \pm 0.1$ | $47.1 \pm 0.3$ | $56.6 \pm 2.0$ | $1487 \pm 68$ | — | — | $16.4 \pm 0.3$ | $46.6 \pm 0.8$ |
| Squared error | — | $\mathbf{44.3 \pm 0.4}$ | — | $1258 \pm 49$ | – | $\mathbf{5.2e4 \pm 8.3e3}$ | — | $\mathbf{32.3 \pm 0.1}$ |

## 5.2 PixelCNN pixel regression

For image modeling tasks, we evaluate the distribution functions on MNIST, FashionMNIST, and CIFAR10 datasets. Results are presented in Tables 4 and 5 for single and mixture models respectively, and Tables 6 and 7 for FID scores.

### 5.2.1 Bits and RMSE

Dalap demonstrates strong performance across all image datasets. In the non-mixture setting (Table 4), Dalap achieves the best bits per dimension on MNIST (0.61) and FashionMNIST (1.23), and performs competitively on CIFAR10 (3.11 bits per dimension), only marginally behind Laplace (3.09). The small difference of 0.02 bits per dimension on CIFAR10 is negligible in practice, especially considering Dalap's

Table 3: Test set results for mixture models on bicycles, upvotes, migration and maestro. Seeded results reported as mean $\pm$ SE over 10 seeds; best in bold.

| | Bicycles | | | Upvotes | | | Migration | | | MAESTRO | |
|---|---|---|---|---|---|---|---|---|---|---|---|
| | K | Bits | RMSE | K | Bits | RMSE | K | Bits | RMSE | K | Bits | RMSE |
| Dalap | 4 | **6.78 ± 0.02** | 130 ± 1 | 8 | **6.49 ± 0.00** | **122 ± 3** | 8 | 20.4 ± 1.0 | 6.2e4 ± 2.4e4 | 8 | 4.88 ± 0.04 | 54.3 ± 0.3 |
| Bitwise | 8 | 8.09 ± 0.03 | 1.1e4 ± 2.1e3 | 4 | 6.93 ± 0.02 | 1.3e6 ± 6.8e5 | 8 | **18.0 ± 0.0** | 9.2e6 ± 2.1e6 | 8 | 5.19 ± 0.02 | 95.5 ± 28.6 |
| Dlaplace | 2 | 6.86 ± 0.02 | 65.9 ± 0.9 | 8 | 6.50 ± 0.01 | 267 ± 6 | 2 | 23.8 ± 4.7 | **3.4e4 ± 1.5e4** | 8 | 4.90 ± 0.00 | **53.9 ± 0.1** |
| Dnormal | 2 | 6.84 ± 0.01 | 66.0 ± 0.9 | 8 | 6.51 ± 0.00 | 543 ± 29 | 8 | 20.0 ± 0.6 | 7.6e4 ± 1.8e4 | 8 | 4.95 ± 0.00 | 57.2 ± 0.2 |
| Dweib | 8 | 7.13 ± 0.03 | 118 ± 3 | 8 | 6.54 ± 0.01 | 1336 ± 141 | — | — | — | 8 | 4.92 ± 0.02 | 56.3 ± 0.3 |
| Danorm | 8 | 7.11 ± 0.22 | 154 ± 33 | 4 | 6.71 ± 0.06 | 1346 ± 104 | 8 | 2807 ± 218 | 2.6e5 ± 8.7e4 | 8 | **4.85 ± 0.00** | 57.0 ± 0.7 |
| Poisson | 8 | 7.41 ± 0.07 | **59.0 ± 0.9** | 8 | 10.9 ± 0.5 | 1711 ± 117 | — | — | — | 8 | 5.29 ± 0.01 | 67.0 ± 0.9 |

superior performance on the other datasets. Dlogistic, included here as the established baseline from PixelCNN++, achieves 0.69 on MNIST and 3.18 on CIFAR10 in the non-mixture setting, placing it behind Dalap on all three datasets. The categorical baseline achieves competitive FID scores for reconstruction (see Section 5.2.2), but exhibits substantially worse bits per dimension compared to distributions that respect ordinal structure (e.g., 1.18 vs. 0.61 for Dalap on MNIST, and 4.55 vs. 3.11 on CIFAR10). This confirms that while a categorical output can produce visually plausible reconstructions, it assigns probability mass less efficiently by treating each intensity value as an unrelated class.

In the mixture setting (Table 5), Dalap maintains its strong performance with 0.66 bits per dimension on MNIST, matching FashionMNIST performance at 1.23, and achieving the best result on CIFAR10 (3.0206 bits per dimension). When comparing against Dlogistic, Dalap outperforms it on CIFAR10 by 0.0027 bits per dimension, and shows comparable performance on MNIST (0.66 vs 0.67). This demonstrates that Dalap can match or exceed the performance of the established discretized continuous distribution.

Examining RMSE scores, Dalap consistently achieves the lowest values in both single and mixture settings across all datasets. In the mixture models, Dalap achieves RMSEs of 330.48 (MNIST), 317.12 (FashionMNIST), and 510.64 (CIFAR10). This indicates that Dalap not only models the probability distribution well but also produces accurate point predictions.

Bitwise shows significantly worse performance on image data, with bits per dimension more than double that of Dalap in most cases. This is particularly pronounced in the mixture setting, where Bitwise achieves 8.02 bits per dimension on CIFAR10 compared to Dalap's 3.02. The high RMSE values for Bitwise (e.g., 5051.66 on CIFAR10 mixture models) further confirm that representing pixels bit-by-bit introduces substantial prediction errors.

The discretized continuous distributions (Laplace and Dnormal) show reasonable performance but generally fall short of Dalap. Laplace performs well on CIFAR10 in both settings but struggles on MNIST (24.47 bits per dimension in the non-mixture case), suggesting that simple discretization may not capture the distribution effectively for all image types. Dnormal exhibits similar patterns, with particularly poor performance on grayscale images.

Table 4: Test set results for image modeling on MNIST, FashionMNIST and CIFAR10, best in bold.

| | MNIST | | FashionMNIST | | CIFAR10 | |
|---|---|---|---|---|---|---|
| | Bits/dim | RMSE | Bits/dim | RMSE | Bits/dim | RMSE |
| Dalap | **0.61** | **253** | **1.23** | **317** | 3.11 | **480** |
| Bitwise | 1.63 | 535 | 3.76 | 881 | 5.66 | 1 534 |
| Laplace | 24.47 | 453 | 5.84 | 383 | **3.09** | 527 |
| Dnormal | 9.10 | 3 992 | 7.91 | 3 711 | 3.37 | 530 |
| Dlogistic | 0.69 | 337 | 1.45 | 485 | 3.18 | 570 |
| Cat | 1.18 | 351 | 2.83 | 489 | 4.55 | 810 |

Table 5: Test set results for mixture models with a K of 10 on MNIST, FashionMNIST and CIFAR10, best in bold.

| | MNIST | | FashionMNIST | | CIFAR10 | |
|---|---|---|---|---|---|---|
| | Bits/dim | RMSE | Bits/dim | RMSE | Bits/dim | RMSE |
| Dalap | **0.66** | **330** | **1.23** | **317** | **3.02** | **511** |
| Bitwise | 2.71 | 6 753 | 4.98 | 5 699 | 8.02 | 5 052 |
| Laplace | 3.43 | 1 085 | 2.20 | 3 539 | 3.05 | 515 |
| Dnormal | 3.97 | 1 643 | 3.09 | 1 111 | 3.07 | 516 |
| Dlogistic | 0.67 | 402 | 1.37 | 555 | **3.02** | 555 |

### 5.2.2 Fréchet Inception Distance

In the FID scores of generated samples (Tables 6 and 7), we observe nuanced performance patterns that depend on both the dataset and the specific generation task.

In the non-mixture setting (Table 6), Dalap achieves the best seeded sampling performance on MNIST (16.12) and FashionMNIST (52.08), demonstrating its ability to generate high-quality samples when conditioned on partial input. For CIFAR10, Laplace achieves the best seeded sampling score (45.74), closely followed by Dalap (49.76). For random sampling, Dalap excels on MNIST (33.19), while Laplace performs best on CIFAR10 (115.11). Interestingly, both the categorical distribution and Dlogistic achieve highly competitive reconstruction FID scores across the datasets. In particular, Dlogistic obtains the best reconstruction FID on FashionMNIST and CIFAR10, while the categorical distribution obtains the best reconstruction FID on MNIST. This highlights a compelling divergence, indicating that a model's log-likelihood does not necessarily correlate perfectly with its perceptual generation quality.

The mixture setting (Table 7) reveals a different pattern. Dlogistic dominates reconstruction tasks across all datasets, achieving FID scores of 42.00 (MNIST), 37.75 (FashionMNIST), and 26.50 (CIFAR10). This higher reconstruction performance likely stems from the smoothness properties of the underlying continuous distribution, which may better capture the perceptual structure of images. However, for sampling tasks, the performance is more balanced. Dalap achieves the best random sampling scores on MNIST (64.75) and FashionMNIST (171.59), while Dlogistic performs best on CIFAR10 (101.71). For seeded sampling, Dlogistic excels on MNIST (18.80), Dalap on FashionMNIST (62.56), and CIFAR10 shows competitive performance across multiple distributions with Laplace (41.12), Dnormal (42.12), and Dalap (44.73) all achieving similar scores.

These results indicate that the choice of distribution for image generation depends on the specific application requirements. Dlogistic offers better reconstruction quality, making it preferable for tasks like image compression or autoencoding where faithful reproduction is most important. However, Dalap demonstrates competitive or better performance for sampling tasks, particularly on grayscale datasets (MNIST, Fashion-

MNIST), while maintaining better likelihood scores (bits per dimension) as shown in Tables 4 and 5. The consistently strong performance of Dalap across both likelihood estimation and generative quality metrics suggests it is a robust choice for pixel-level modeling tasks where both probabilistic modeling and sample quality matter.

Bitwise, despite its competitive reconstruction FID in the non-mixture setting, shows poor performance in mixture models and across all sampling tasks. Dnormal exhibits mixed performance, occasionally achieving competitive scores (e.g., 42.12 for CIFAR10 seeded sampling in a mixture setting) but lacking consistency across tasks and datasets.

Table 6: FID scores on test set for image modeling on MNIST, FashionMNIST and CIFAR10, best in bold.

|          | MNIST | | | FashionMNIST | | | CIFAR10 | | |
|----------|-------|--------|--------|--------|--------|--------|--------|--------|--------|
|          | Recon. | Seeded | Random | Recon. | Seeded | Random | Recon. | Seeded | Random |
| Dalap    | 230   | **16** | **33** | 236 | 52 | 154 | 308 | 50 | 146 |
| Bitwise  | 66    | 56     | 90     | 106 | 145 | 299 | 120 | 211 | 290 |
| Laplace  | 145   | 109    | 107    | 258 | 102 | 226 | 308 | **46** | **115** |
| Dnormal  | 438   | 274    | 354    | 372 | 268 | 381 | 294 | 67 | 146 |
| Dlogistic| 41    | **16** | 44     | **33** | 69 | 214 | **36** | 70 | 133 |
| Cat      | **39**| 40     | 65     | 41 | **47** | **129** | 45 | 160 | 258 |

Table 7: FID scores for mixture models with a K of 10 on test set for image modeling on MNIST, Fashion-MNIST and CIFAR10, best in bold.

|          | MNIST | | | FashionMNIST | | | CIFAR10 | | |
|----------|-------|--------|--------|--------|--------|--------|--------|--------|--------|
|          | Recon. | Seeded | Random | Recon. | Seeded | Random | Recon. | Seeded | Random |
| Dalap    | 232   | 54     | **65** | 258 | **63** | **172** | 306 | 45 | 126 |
| Bitwise  | 316   | 287    | 372    | 296 | 275 | 397 | 394 | 264 | 407 |
| Laplace  | 232   | 155    | 186    | 304 | 207 | 389 | 306 | **41** | 138 |
| Dnormal  | 249   | 180    | 217    | 306 | 146 | 218 | 308 | 42 | 189 |
| Dlogistic| **42**| **19** | 90     | **38** | 69 | 200 | **27** | **41** | **102** |

Overall, these experiments demonstrate that Dalap offers strong performance for discrete pixel prediction, matching or exceeding the established discretized logistic mixture approach in likelihood estimation while maintaining competitive generative quality, particularly for sampling tasks.

### 5.2.3   Generated samples

We present visual samples in Figures 3, 4 and 5. These figures show results from the mixture models (K=10) which provide the best results. Additional samples for all evaluated distributions are provided in Appendix A.2.

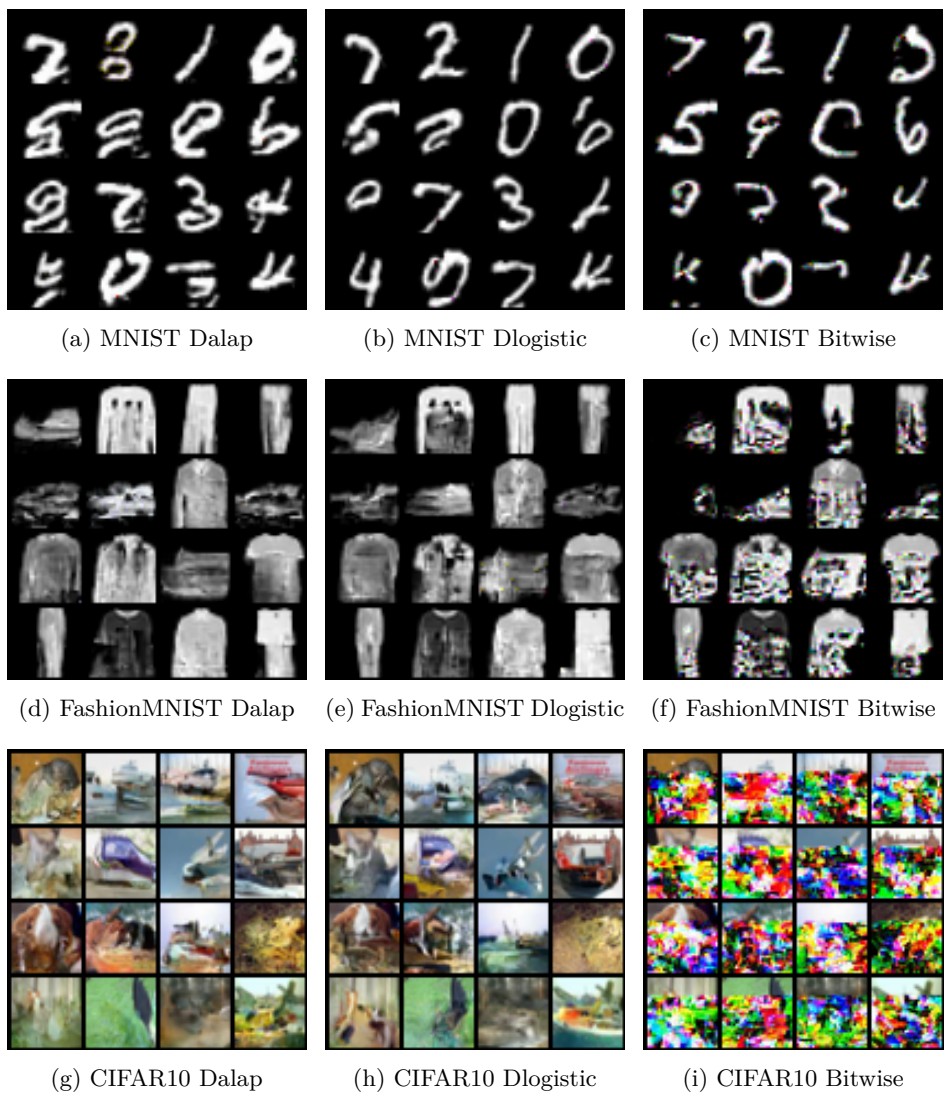

Figure 3: Seeded sampling results from mixture models ($K = 10$). The top portion of each image is provided as conditioning, and the model generates the remainder. Dalap produces high-quality completions across all datasets, with visual performance comparable to Dlogistic (PixelCNN++) and competitive or lower bits-per-dimension scores. Bitwise exhibits visible artifacts, particularly on CIFAR10, consistent with its poor quantitative performance.

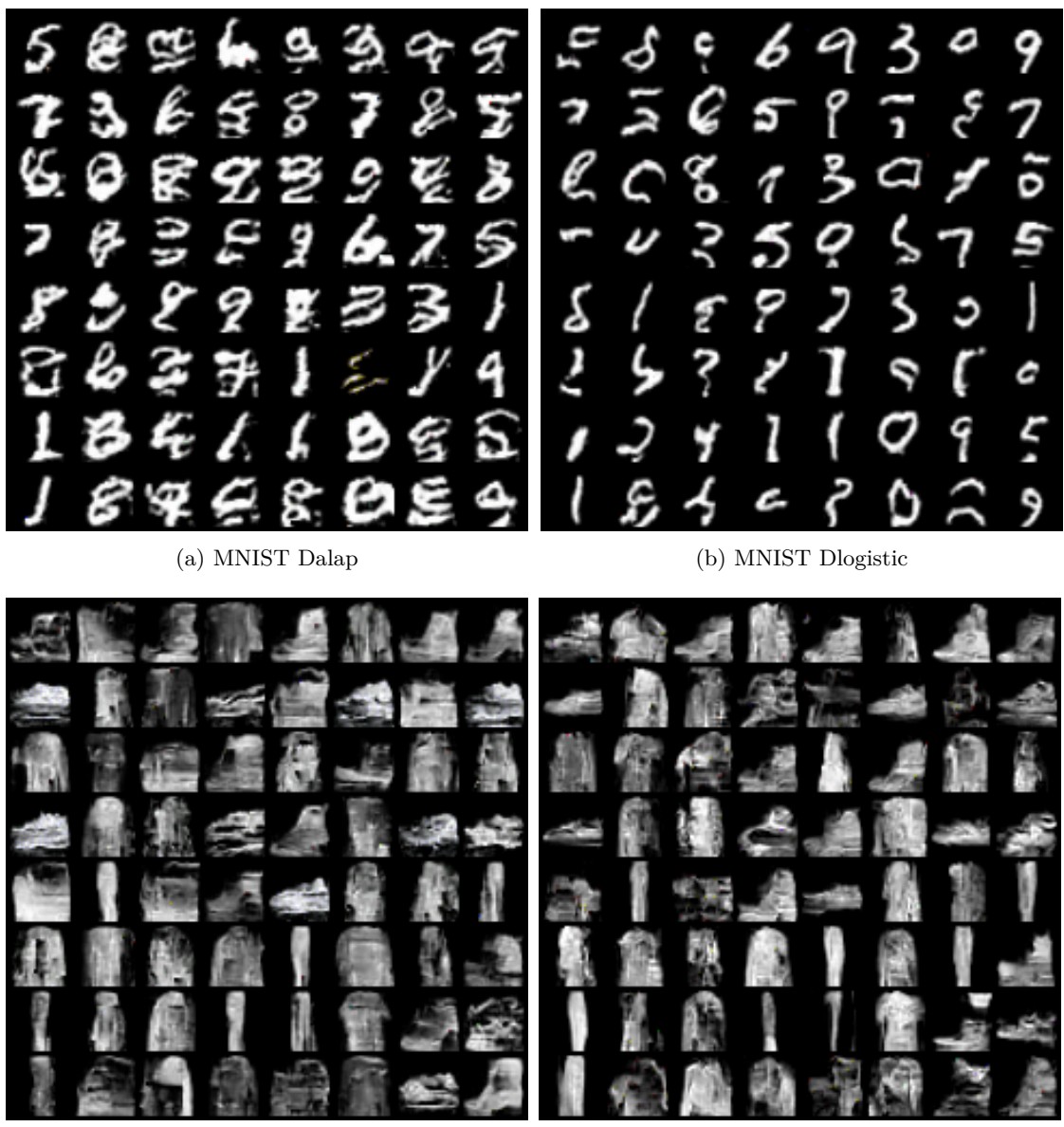

(a) MNIST Dalap

(b) MNIST Dlogistic

(c) FashionMNIST Dalap

(d) FashionMNIST Dlogistic

Figure 4: Random (unconditional) sampling results on MNIST and FashionMNIST from mixture models (K=10). All images are generated from scratch without any conditioning.

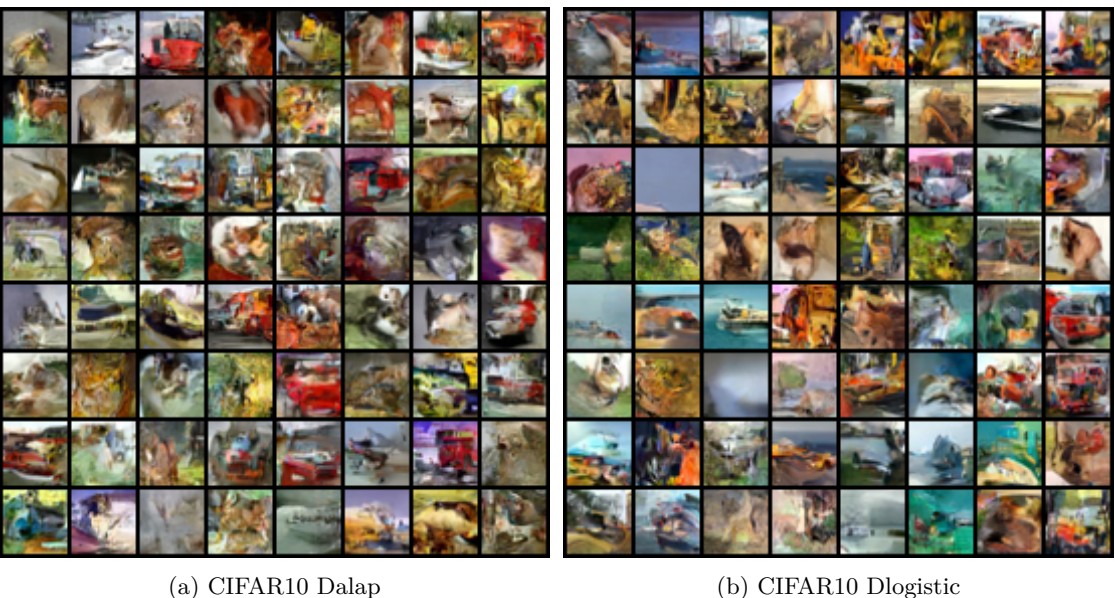

(a) CIFAR10 Dalap                    (b) CIFAR10 Dlogistic

Figure 5: Random (unconditional) sampling results on CIFAR10 from mixture models (K=10). All images are generated from scratch without any conditioning.

# 6 Conclusion

In this work, we investigated the problem of predicting integer labels from continuous parameters, specifically within the context of neural networks where continuous weights are required for backpropagation. We evaluated several discrete distributions across tabular regression, sequential prediction, and image generation tasks, comparing them against established baselines including the Poisson distribution, categorical distribution, and the discretized logistic mixture used in PixelCNN++.

Our results demonstrate that modeling integer data with truly discrete distributions can be competitive with continuous relaxations. While continuous relaxations, optimizing for squared error, generally yield the lowest RMSE in tabular settings, they lack the ability to provide valid probability mass functions on the integers. Similarly, while the Poisson baseline performs poorly in terms of bits across all datasets, confirming its single-parameter form cannot capture complex variance structures, it still achieves competitive RMSE on datasets like Bicycles and MAESTRO. When a discrete distribution is required, for instance to minimize compression loss (bits), we found that the Discrete Analogue of the Laplace distribution (Dalap) offers the strongest overall performance across the settings we evaluated.

Specifically, Dalap consistently achieves the best or near-best results in terms of negative log-likelihood on the tabular and image datasets, matching or exceeding the performance of the discretized logistic mixture used in PixelCNN++. It provides a robust balance between minimizing bits and maintaining low RMSE, and offers a more mathematically principled approach to modeling pixel value distributions, which are inherently discrete. In contrast, the discretized logistic relies entirely on continuous computations and only discretizes through rounding at the final prediction stage. That said, no single distribution dominates in all settings: on the highly dispersed Migration dataset, Dnormal and Bitwise outperform Dalap in bits, suggesting that the choice of distribution should ultimately be informed by the characteristics of the target data.

The use of mixture models proved important for handling challenging data. On the Migration dataset, Dalap without mixture components showed sensitivity to initialization (2 of 10 seeds diverged), but this instability was resolved with $K = 8$ components, yielding stable convergence across all seeds. More broadly, mixture models improved both bits and RMSE for most distributions across most datasets.

The Bitwise distribution proves to be a valuable alternative for high-entropy, highly dispersed data, achieving the best bits on Migration in the mixture setting. However, it exhibits a high sensitivity to uncertainty in higher-order bits. If the model is uncertain about significant bits, the prediction can deviate substantially from the target, resulting in the high variance and RMSE observed in our experiments. The categorical distribution, evaluated on the image datasets, achieves competitive reconstruction quality (FID) but substantially worse bits per dimension, confirming the importance of respecting ordinal structure in integer-valued data.

Ultimately, we conclude that continuous relaxations remain strong baselines for pure error minimization, especially in the non-mixture setting, but they are not uniformly superior once discrete mixture models are considered. When a discrete probabilistic model is necessary, Dalap is the preferred choice across the settings we evaluated. It offers the strongest overall balance of distributional quality (bits) and prediction accuracy (RMSE), while Bitwise and Dnormal may be preferable for data with extreme dispersion characteristics.

## 6.1 Limitations and future work

Our research faced limitations primarily driven by computational constraints. Due to limitations with time and resources, we were unable to perform an exhaustive grid search for the optimal number of mixture components $K$ for the image modeling experiments. It is possible that fine-tuning this hyperparameter could yield further performance improvements for the mixture models. Because we compared our distribution functions to Dlogistic using the PixelCNN++ architecture, we adopted the hyperparameters from their best-performing results. Furthermore, the image experiments were conducted with a single random seed due to the high computational cost of training PixelCNN models (each requiring multiple days on a single GPU), which limits the conclusions that can be drawn about the variability of results in that setting.

Additionally, while theoretically sound, some distributions proved impractical for high-dimensional tasks. The Discrete Analogue of the Normal distribution (Danorm) requires a numerical approximation of the par-

tition function that scales poorly with input size. In our image generation experiments, Danorm required over 90GB of VRAM, exceeding available hardware capacity and forcing its exclusion from those benchmarks. Similarly, the Discrete Weibull (Dweib) distribution exhibited numerical instability, particularly when modeling bounded intervals, leading to convergence failures. It may be, however, that with alternative implementations or additional tricks to stabilize learning, these distributions can still be made to work well.

Future work will focus on addressing these stability and efficiency issues. Furthermore, while we demonstrated the efficacy of these distributions on the specific sub-task of predicting note duration in ticks, a natural next step is to integrate Dalap into a full-scale autoregressive generative model for MIDI music, moving beyond conditional scalar prediction to full sequence generation.

### Author Contributions

Both authors contributed equally to ideation, coding and writing of the article.

### Acknowledgments

We used the DAS-6 Bal et al. (2016) infrastructure at Vrije Universiteit Amsterdam, funded by NWO and participating universities, as well as SURF (www.surf.nl) under the grant EINF-5362.

## References

Agriculture & Rural Development. World Bank staff estimates based on the United Nations Population Division's World Urbanization Prospects: 2018 Revision, 2018. Folder IDs SP.RUR.TOTL.ZS, SP.RUR.TOTL.ZG, SP.RUR.TOTL, AG.SRF.TOTL.K2, ISAD(G) Reference Code World Urbanization Prospects: 2018 Revision, [Each Level Label as applicable], World Bank Group Archives, Washington, D.C., United States.

Net Migration Data. United Nations Population Division. World Population Prospects: 2022 Revision, 2022. Net migration, Folder ID SM.POP.NETM, ISAD(G) Reference Code World Population Prospects: 2022 Revision, [Each Level Label as applicable], World Bank Group Archives, Washington, D.C., United States.

Climate Change Indicators. World Bank staff estimates and other relevant data sources, 2024. Folder IDs SP.URB.TOTL.IN.ZS, SP.URB.TOTL, SP.POP.TOTL, SP.POP.GROW, ISAD(G) Reference Code Climate Change Statistics and other relevant sources, [Each Level Label as applicable], World Bank Group Archives, Washington, D.C., United States.

Environment Indicators. World Bank staff estimates and other relevant data sources, 2024. Folder IDs ER.FSH.PROD.MT, ER.FSH.CAPT.MT, ER.FSH.AQUA.MT, ISAD(G) Reference Code International Environmental Statistics and other relevant sources, [Each Level Label as applicable], World Bank Group Archives, Washington, D.C., United States.

Financial Sector. World Bank staff estimates and other relevant data sources, 2024. Folder IDs SM.POP.NETM, PA.NUS.FCRF, PA.NUS.ATLS, BX.KLT.DINV.CD.WD, ISAD(G) Reference Code International Financial Statistics and other relevant sources, [Each Level Label as applicable], World Bank Group Archives, Washington, D.C., United States.

Ayman Alzaatreh, Carl Lee, and Felix Famoye. On the discrete analogues of continuous distributions. *Statistical Methodology*, 9(6):589–603, 2012.

H. Bal, D. Epema, C. de Laat, R. van Nieuwpoort, J. Romein, F. Seinstra, C. Snoek, and H. Wijshoff. A medium-scale distributed system for computer science research: Infrastructure for the long term. *Computer*, 49(05):54–63, may 2016. ISSN 1558-0814. doi: 10.1109/MC.2016.127.

Wagner H Bonat, Bent Jørgensen, Célestin C Kokonendji, John Hinde, and Clarice GB Demétrio. Extended poisson–tweedie: properties and regression models for count data. *Statistical Modelling*, 18(1):24–49, 2018.

A Colin Cameron and Per Johansson. Count data regression using series expansions: with applications. *Journal of Applied Econometrics*, 12(3):203–223, 1997.

A. Colin Cameron and Pravin K. Trivedi. *Regression Analysis of Count Data*, volume 53. Cambridge University Press, 2013.

Subrata Chakraborty. Generating discrete analogues of continuous probability distributions-a survey of methods and constructions. *Journal of Statistical Distributions and Applications*, 2:1–30, 2015.

Kipkorir Collins, Anthony Waititu, and Anthony Wanjoya. Discrete weibull and artificial neural network models in modelling over-dispersed count data. *Int J Data Sci Anal*, 6(5):153–62, 2020.

Thomas M Cover. *Elements of information theory*. John Wiley & Sons, 1999.

Prafulla Dhariwal and Alexander Nichol. Diffusion models beat gans on image synthesis. *Advances in neural information processing systems*, 34:8780–8794, 2021.

Thomas G. Dietterich and Ghulum Bakiri. Solving multiclass learning problems via error-correcting output codes. *J. Artif. Intell. Res.*, 2:263–286, 1995. doi: 10.1613/JAIR.105. URL https://doi.org/10.1613/jair.105.

Mahmoud El-Morshedy, Mohammed S Eliwa, and Emrah Altun. Discrete burr-hatke distribution with properties, estimation methods and regression model. *IEEE access*, 8:74359–74370, 2020.

James D Englehardt and Ruochen Li. The discrete weibull distribution: An alternative for correlated counts with confirmation for microbial counts in water. *Risk Analysis: An International Journal*, 31(3):370–381, 2011.

Hadi Fanaee-T. Bike Sharing. UCI Machine Learning Repository, 2013. DOI: https://doi.org/10.24432/C5W894.

Emilio Gómez-Déniz. Another generalization of the geometric distribution. *Test*, 19:399–415, 2010.

Emilio Gómez-Déniz and Enrique Calderín-Ojeda. The discrete lindley distribution: properties and applications. *Journal of statistical computation and simulation*, 81(11):1405–1416, 2011.

Peter D Grünwald. *The minimum description length principle*. MIT press, 2007.

Curtis Hawthorne, Andriy Stasyuk, Adam Roberts, Ian Simon, Cheng-Zhi Anna Huang, Sander Dieleman, Erich Elsen, Jesse Engel, and Douglas Eck. Enabling factorized piano music modeling and generation with the MAESTRO dataset. In *International Conference on Learning Representations*, 2019. URL https://openreview.net/forum?id=r1lYRjC9F7.

Joseph M Hilbe. *Modeling count data*. Cambridge University Press, 2014.

Jonathan Ho, Ajay Jain, and Pieter Abbeel. Denoising diffusion probabilistic models. *Advances in neural information processing systems*, 33:6840–6851, 2020.

Seidu Inusah and Tomasz J Kozubowski. A discrete analogue of the laplace distribution. *Journal of statistical planning and inference*, 136(3):1090–1102, 2006.

Adrienne W Kemp. Characterizations of a discrete normal distribution. *Journal of Statistical Planning and Inference*, 63(2):223–229, 1997.

Alex Krizhevsky and Geoffrey Hinton. Learning multiple layers of features from tiny images. Technical report, University of Toronto, 2009.

Yann LeCun, Léon Bottou, Yoshua Bengio, and Patrick Haffner. Gradient-based learning applied to document recognition. *Proceedings of the IEEE*, 86(11):2278–2324, 1998.

Gabriel Loaiza-Ganem and John P. Cunningham. The continuous bernoulli: fixing a pervasive error in variational autoencoders, 2019. URL https://arxiv.org/abs/1907.06845.

Toshio Nakagawa and Shunji Osaki. The discrete weibull distribution. *IEEE Transactions on Reliability*, R-24(5):300–301, 1975. doi: 10.1109/TR.1975.5214915.

Umair Naseer. Predict the number of upvotes a post will get. `https://www.kaggle.com/datasets/umairnsr87/predict-the-number-of-upvotes-a-post-will-get`, 2020. Accessed: March 20, 2024.

Vahid Nekoukhou and Hamid Bidram. The exponentiated discrete weibull distribution. *Sort*, 39:127–146, 2015.

Alec Radford, Jeffrey Wu, Rewon Child, David Luan, Dario Amodei, Ilya Sutskever, et al. Language models are unsupervised multitask learners. *OpenAI blog*, 1(8):9, 2019.

Aditya Ramesh, Mikhail Pavlov, Gabriel Goh, Scott Gray, Chelsea Voss, Alec Radford, Mark Chen, and Ilya Sutskever. Zero-shot text-to-image generation, 2021. URL `https://arxiv.org/abs/2102.12092`.

Dilip Roy. Discrete rayleigh distribution. *IEEE transactions on reliability*, 53(2):255–260, 2004.

Oleh Rybkin, Kostas Daniilidis, and Sergey Levine. Simple and effective vae training with calibrated decoders. In *International conference on machine learning*, pp. 9179–9189. PMLR, 2021.

Tim Salimans, Andrej Karpathy, Xi Chen, and Diederik P. Kingma. Pixelcnn++: Improving the pixelcnn with discretized logistic mixture likelihood and other modifications, 2017. URL `https://arxiv.org/abs/1701.05517`.

Terrence J. Sejnowski and Charles R. Rosenberg. Nettalk: a parallel network that learns to read aloud. *The Johns Hopkins University Electrical Engineering and Computer Science Technical Report*, 1986.

Galit Shmueli, Thomas P Minka, Joseph B Kadane, Sharad Borle, and Peter Boatwright. A useful distribution for fitting discrete data: revival of the conway–maxwell–poisson distribution. *Journal of the Royal Statistical Society Series C: Applied Statistics*, 54(1):127–142, 2005.

Chénangnon Frédéric Tovissodé, Sèwanou Hermann Honfo, Jonas Têlé Doumatè, and Romain Glèlè Kakaï. On the discretization of continuous probability distributions using a probabilistic rounding mechanism. *Mathematics*, 9(5):555, 2021.

Aaron van den Oord, Nal Kalchbrenner, Oriol Vinyals, Lasse Espeholt, Alex Graves, and Koray Kavukcuoglu. Conditional image generation with pixelcnn decoders, 2016. URL `https://arxiv.org/abs/1606.05328`.

World Bank. World bank website. `https://www.worldbank.org`, 2024. Accessed: 2024-09-01.

Han Xiao, Kashif Rasul, and Roland Vollgraf. Fashion-mnist: a novel image dataset for benchmarking machine learning algorithms. *arXiv preprint arXiv:1708.07747*, 2017.

# A  Appendix

## A.1  Exploratory analysis of the target distributions

In this appendix we summarize the empirical characteristics of the target distributions of every dataset used in the paper. The goal is to make explicit why no single distribution function is preferred across all benchmarks: the datasets span a wide range of supports, dispersion regimes and tail behaviours, and a candidate distribution that fits one regime well does not necessarily fit another.

Table 8 reports the support, sample size, range, mean, variance, dispersion index $\sigma^2/\mu$, skewness and excess kurtosis for each target. Bicycles is mildly overdispersed, while Upvotes and Migration are extremely overdispersed and heavy tailed. MAESTRO has a moderate dispersion index, which is consistent with the relative success of the Poisson baseline on this dataset (Section 5.1). The pixel targets of MNIST, FashionMNIST and CIFAR10 occupy the bounded support $\{0, \ldots, 255\}$ and are dominated by extreme values, with most of their mass concentrated near 0 and 255 for MNIST and FashionMNIST and a broader but still bimodal shape for CIFAR10.

Table 8: Summary statistics for the target distributions of each dataset. $N$ is the number of samples; DI is the dispersion index $\sigma^2/\mu$; skew and kurt are sample skewness and excess kurtosis.

| Dataset | Support | $N$ | min | max | $\mu$ | $\sigma^2$ | DI | skew | kurt |
|---|---|---|---|---|---|---|---|---|---|
| Bicycles | $\mathbb{Z}_{\geq 0}$ | 17 379 | 1 | 977 | 189.46 | $3.29 \times 10^4$ | 173.66 | 1.28 | 1.42 |
| Upvotes | $\mathbb{Z}_{\geq 0}$ | 330 045 | 0 | 615 278 | 337.51 | $1.29 \times 10^7$ | $3.82 \times 10^4$ | 74.25 | 8919.66 |
| Migration | $\mathbb{Z}$ | 7 945 | $-2\,290\,411$ | 1 866 819 | 1601.57 | $2.96 \times 10^{10}$ | $1.85 \times 10^7$ | 0.70 | 50.21 |
| MAESTRO | $\mathbb{Z}_{\geq 0}$ | 19 826 496 | 0 | 18 803 | 21.57 | 3235.24 | 149.99 | 47.58 | 7149.41 |
| MNIST | $\{0, \ldots, 255\}$ | 7 840 000 | 0 | 255 | 33.41 | 6186.76 | 185.17 | 2.15 | 2.89 |
| FashionMNIST | $\{0, \ldots, 255\}$ | 7 840 000 | 0 | 255 | 72.64 | 8045.79 | 110.77 | 0.70 | $-1.18$ |
| CIFAR10 | $\{0, \ldots, 255\}$ | 30 720 000 | 0 | 255 | 120.38 | 4124.90 | 34.27 | 0.23 | $-0.79$ |

Figure 6 shows the empirical histogram of each target on a logarithmic count axis. Plotting on a log scale is necessary because the heavy tailed datasets place several orders of magnitude more mass near the mode than in the tail, and a linear axis hides the structure that the discrete distributions in Section 3 are designed to capture. The horizontal axis is clipped to the central 98 percent of the data and the full range is reported in the panel annotations.

Figure 7 shows the dispersion index $\sigma^2/\mu$ for each dataset on a logarithmic scale.

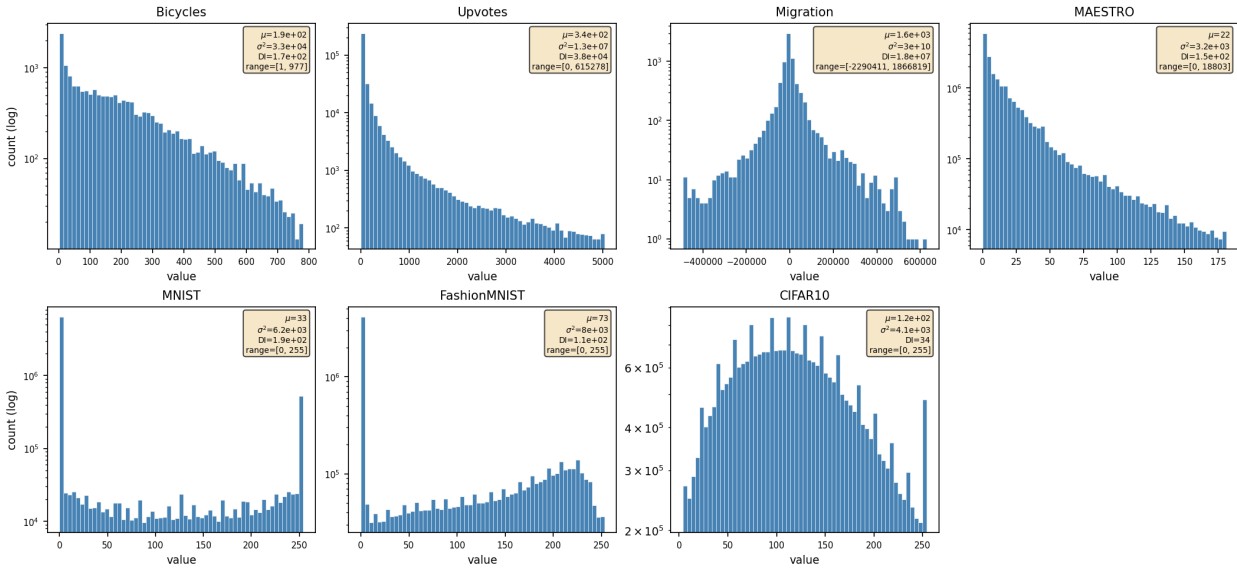

Figure 6: Empirical histograms of the target distributions for every dataset used in the paper. The vertical axis is logarithmic. The annotation in each panel reports the sample mean $\mu$, sample variance $\sigma^2$, dispersion index DI and the full range of the data before clipping.

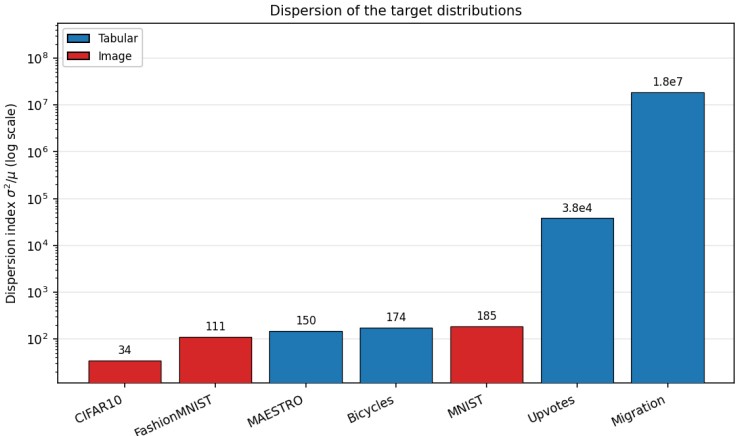

Figure 7: Dispersion index $\sigma^2/\mu$ for each target distribution (log scale). Tabular datasets in blue, image datasets in red.

## A.2   Generated image examples

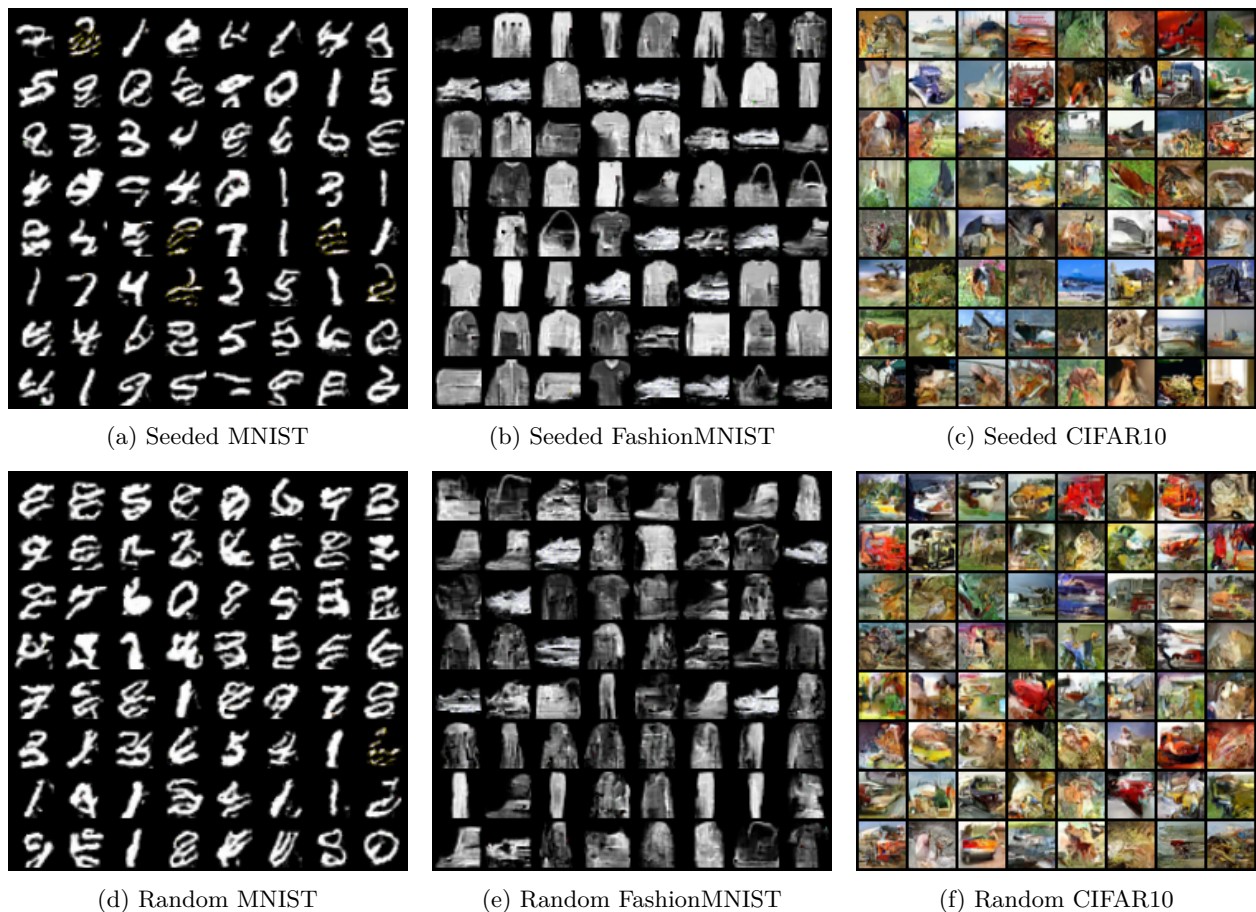

(a) Seeded MNIST                (b) Seeded FashionMNIST                (c) Seeded CIFAR10

(d) Random MNIST               (e) Random FashionMNIST               (f) Random CIFAR10

Figure 8: Dalap mixture model (K=10). Top row: seeded sampling where the top portion of each image is provided as conditioning. Bottom row: unconditional random sampling.

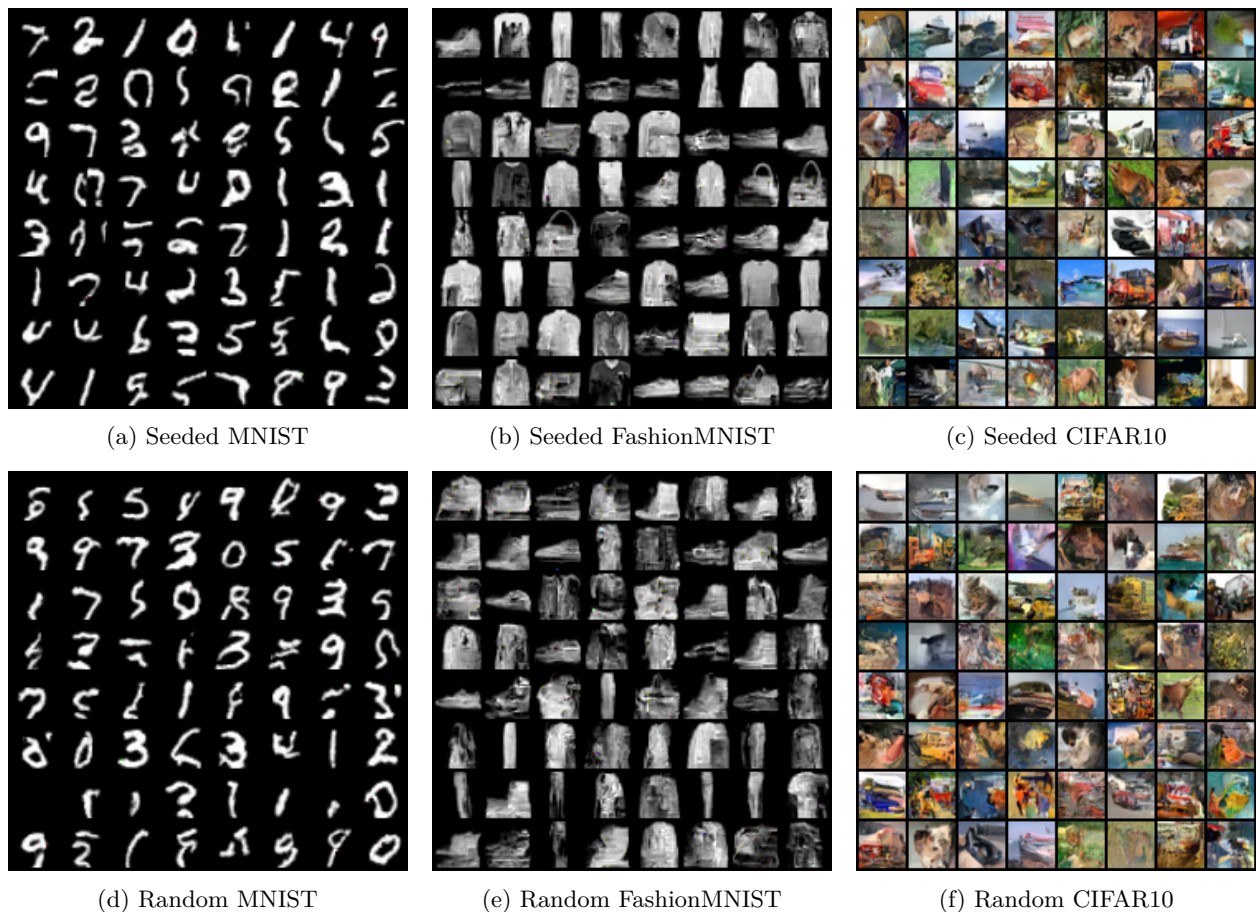

(a) Seeded MNIST    (b) Seeded FashionMNIST    (c) Seeded CIFAR10

(d) Random MNIST    (e) Random FashionMNIST    (f) Random CIFAR10

Figure 9: Dlogistic mixture model (K=10). Top row: seeded sampling where the top portion of each image is provided as conditioning. Bottom row: unconditional random sampling.

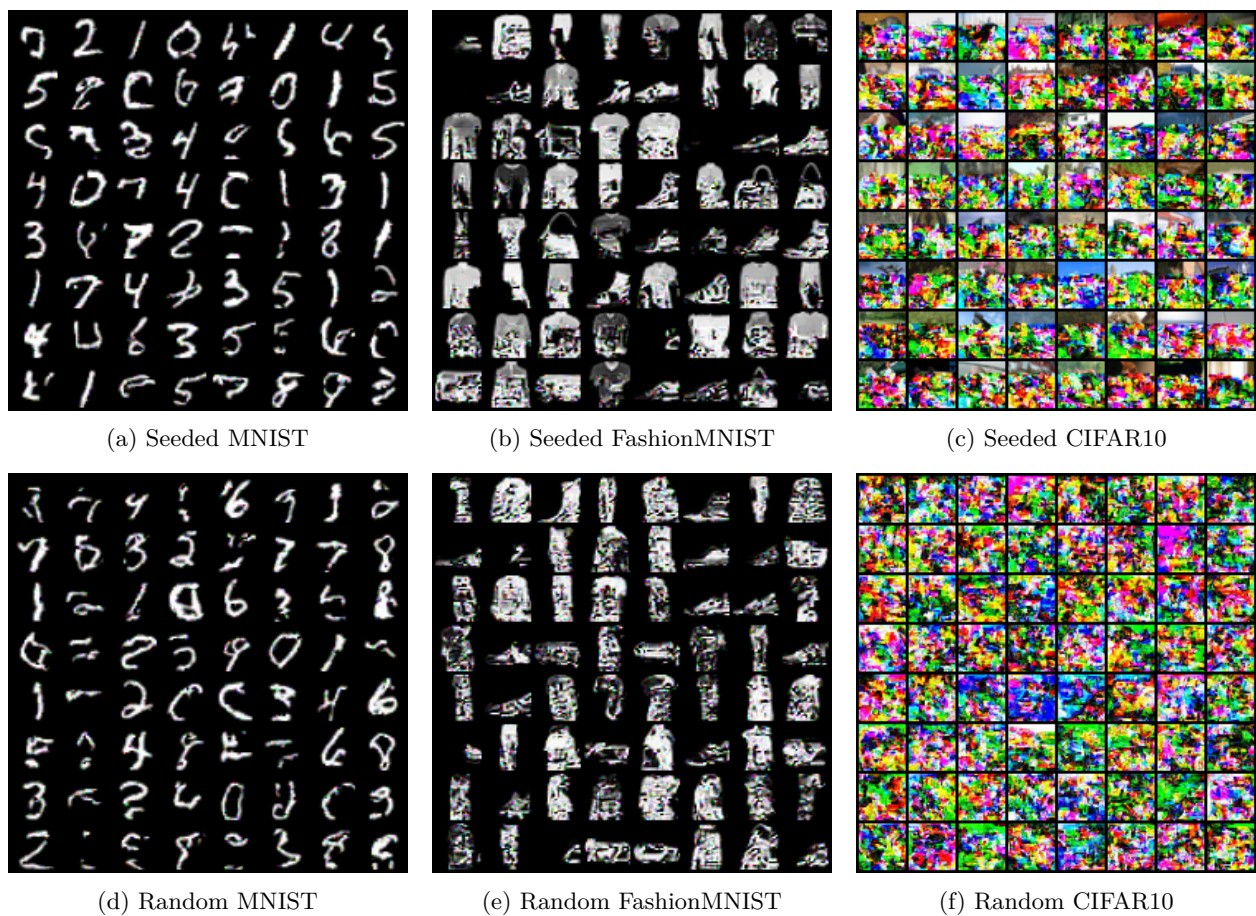

Figure 10: Bitwise model (K=10). Top row: seeded sampling where the top portion of each image is provided as conditioning. Bottom row: unconditional random sampling.

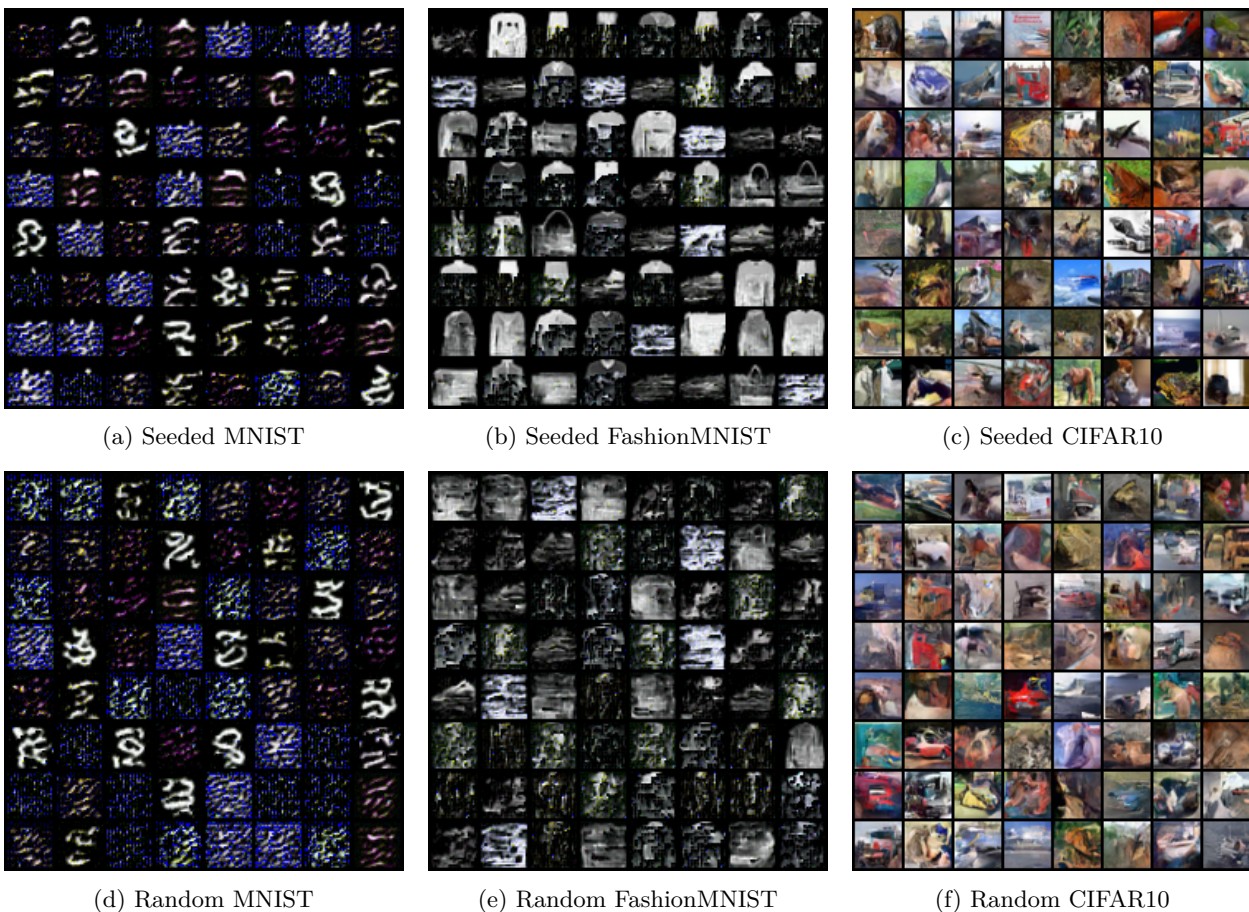

(a) Seeded MNIST    (b) Seeded FashionMNIST    (c) Seeded CIFAR10

(d) Random MNIST    (e) Random FashionMNIST    (f) Random CIFAR10

Figure 11: Dnormal mixture model (K=10). Top row: seeded sampling where the top portion of each image is provided as conditioning. Bottom row: unconditional random sampling.

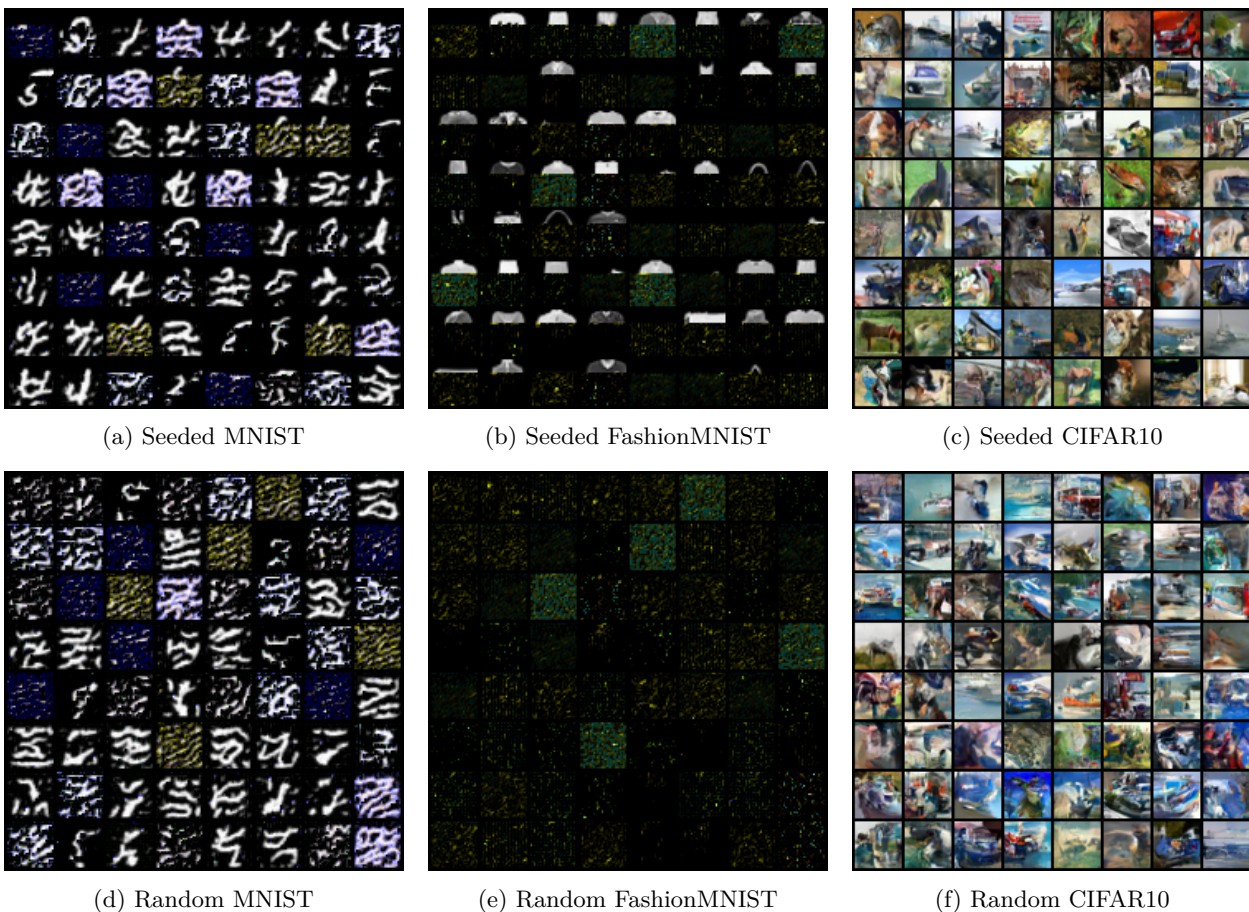

(a) Seeded MNIST      (b) Seeded FashionMNIST      (c) Seeded CIFAR10

(d) Random MNIST      (e) Random FashionMNIST      (f) Random CIFAR10

Figure 12: Laplace mixture model (K=10). Top row: seeded sampling where the top portion of each image is provided as conditioning. Bottom row: unconditional random sampling.

### A.3   Terms of the Dalap loss

In the unbounded case, the term $\log z$ of the Dalap log loss becomes

$$\log z = -\log(1-\gamma) + \log(\gamma^f + \gamma^c).$$

The first term can be interpreted as simply promoting low values of $\gamma$, forcing the model towards high certainty where possible. This term goes to infinity as $\gamma \to 0$, so its effect is substantial, but we find similar terms in the negative log probability of the normal and Laplace distributions. This low variance must be balanced with the exponentially low probability we get for the target $n$ if the distance $|\mu - n|$ is large. [11]

The second term

$$\log(\gamma^c + \gamma^f), \tag{4}$$

when taken in isolation, paints a counter-intuitive picture. As Figure 13 shows, its value becomes a large negative value for non-integers as $\gamma \to 0$. In other words, it seems like a regularization term that rewards non-integer values of $\mu$ under high certainty, even though as our certainty grows, we should be clustered more and more around an integer.

To resolve the apparent contradiction, we must look at the whole loss. Assume first that $d = |\mu - n|$ is some integer value. We will ask what the cost is of moving some small distance $s$ away from $n$ to $d' = d + s$. By Figure 13, we gain a certain amount in the loss due to the term (4). However, the loss is a linear function of the distance which also increases, so we lose something as well.

First, we rewrite (4) as

$$\log(\gamma^c + \gamma^f) = log(\gamma^{1-s} + \gamma^s) = \log(\gamma^{1-2s} + 1) + s\log\gamma.$$

With that the loss becomes:

$$\begin{aligned}
-\log p(n) &= (d+s)\log\frac{1}{\gamma} + \log(\gamma^f + \gamma^c) - \log(1-\gamma) \\
&= -(d+s)\log\gamma + s\log\gamma + \log(1+\gamma^{1-2s}) - \log(1-\gamma) \\
&= -d\log\gamma - s\log\gamma + s\log\gamma + \log(1+\gamma^{1-2s}) - \log(1-\gamma) \\
&= -d\log\gamma + \log(1+\gamma^{1-2s}) - \log(1-\gamma).
\end{aligned}$$

Compare this to the loss we get without moving away from the integer distance:

$$\begin{aligned}
-\log p(n) &= d\log\frac{1}{\gamma} + \log(\gamma^f + \gamma^c) - \log(1-\gamma) \\
&= -d\log\gamma + \log(1+\gamma) - \log(1-\gamma).
\end{aligned}$$

The term in green is the difference. To see how much more we pay in loss for picking the former, we write the difference as

$$\log(1+\gamma^{1-2s}) - \log(1+\gamma) = \log\frac{1+\gamma^{1-2s}}{1+\gamma}.$$

---

[11] In our implementation we ensure in the activation of $\gamma$ that its value does not go below a certain $\epsilon$ to avoid numerical instability.

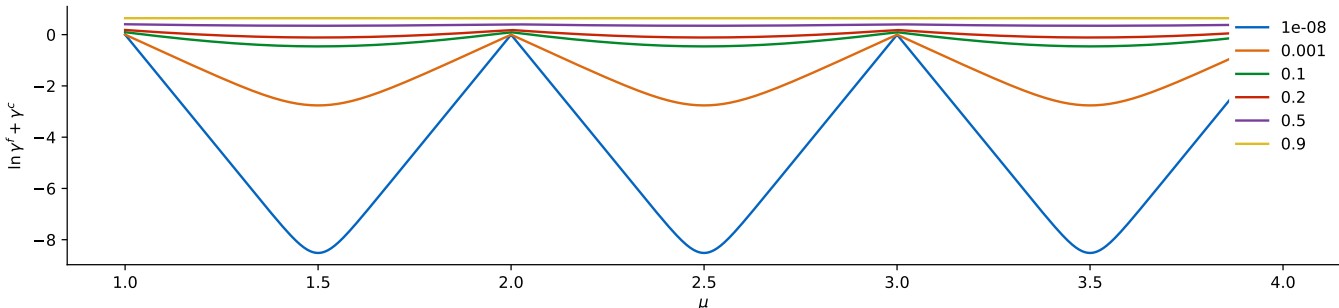

Figure 13: The value of the penalty term $\log(\gamma^f + \gamma^c)$ for different values of $\gamma$. As the value of $\gamma$ goes to 0, non-integer values of $\mu$ appear to receive a benefit.

We then take the limit as $\gamma \to 0$. For $s = 0$, the difference is 0, and for $s = \frac{1}{2}$, the limit is $\log 2$. This tells us that if we move $\mu$ to a non-integer value, we pay a small price in the loss, despite what Figure 13 suggests. This shows that we always pay for increasing the distance to the target.

If we move away from the integer value in the opposite direction (from $d$ to $d - s$) the difference becomes

$$-2s \log \frac{1}{\gamma} + \log \frac{1 + \gamma^{1-2s}}{1 + \gamma}.$$

The first term shows what we gain, and the second what we pay. This shows a balance between improving the distance to the target and staying at integer values of $\mu$. If we let $\gamma \to 0$, then we know that the right term goes to $\log 2$ at most for $s \leq 1/2$ and the left grows unbounded, so any move away from the integer is preferred. For low certainties, say $\gamma = 1/2$, we get

$$-2s \log 2 + \log \left(1 + \left(\frac{1}{2}\right)^{1-2s}\right) - \log \frac{3}{2} = \log \frac{1 + 2^{1-2s}}{3}.$$

As we increase $s$ from 0 to $\frac{1}{2}$, $2^{1-2s}$ decreases monotonically from 2 to 1 and the argument of the logarithm decreases monotonically from 1 to $\frac{1}{3}$, which means the difference function is 0 at $s = 0$ (as expected) and stays below 0 for $s > 0$. This shows that again, moving away from the integer value is always preferred if it decreases the distance to the target.

### A.4 Parameter activations

Each distribution requires its raw network outputs to be mapped to valid parameter ranges. Table 9 summarizes the activation functions used for each support type. Here, $x_1$ and $x_2$ denote the raw network outputs for the location and scale parameters respectively.

The maximum values $\gamma_{\max}$ and $s_{\max}$ are hyperparameters. For Dalap and Danorm, $\gamma_{\max}$ controls the maximum dispersion: values close to 1 allow wide distributions but may cause numerical instability, while lower values constrain the tails. In our tabular and MAESTRO experiments we use $\gamma_{\max} = 1.0$ for Dalap and Danorm, while for the image experiments we use $\gamma_{\max} = 0.9$ to maintain stability. For the discretized distributions, $s_{\max} = 1.0$ in all experiments. These values were found through manual tuning.

### A.5 Numerical stability

**Dalap interval-bounded partition.** The two-way bounded partition function (Section 3.4) involves terms of the form $\gamma^{1+\mu-l}$ and $\gamma^{1+u-\mu}$, which can overflow when $\mu$ is far outside $[l, u]$. To prevent this, we clamp $\mu$ to $[l, u]$ before computing the partition function. For values of $\mu$ outside the interval, we use a

Table 9: Activation functions applied to raw network outputs for each distribution and support type. $\sigma$ denotes the sigmoid function.

| Distribution | Support | Location ($\mu$) | Scale |
|---|---|---|---|
| Dalap | $\mathbb{Z}$ | $x_1$ | clamp($\sigma(x_2) \cdot \gamma_{\max}$, $\epsilon$, $1-\epsilon$) |
|  | $[l, \infty)$ | $\lvert x_1 \rvert + l$ |  |
|  | $[l, u]$ | $\sigma(x_1) \cdot (u - l) + l$ |  |
| Danorm | $\mathbb{Z}$ | $x_1$ | clamp($\sigma(x_2) \cdot \gamma_{\max}$, $\epsilon$, $1-\epsilon$) |
|  | $[l, \infty)$ | $\lvert x_1 \rvert + l$ |  |
|  | $[l, u]$ | $\sigma(x_1) \cdot (u - l) + l$ |  |
| Dnormal | $\mathbb{Z}$ | $x_1$ | softplus($x_2$) $\cdot s_{\max} + \epsilon$ |
|  | $[l, \infty)$ | $\lvert x_1 \rvert + l$ |  |
|  | $[l, u]$ | $\sigma(x_1) \cdot (u - l) + l$ |  |
| Dlaplace | $\mathbb{Z}$ | $x_1$ | softplus($x_2$) $\cdot s_{\max} + \epsilon$ |
|  | $[l, \infty)$ | $\lvert x_1 \rvert + l$ |  |
|  | $[l, u]$ | $\sigma(x_1) \cdot (u - l) + l$ |  |
| Dlogistic | $[l, u]$ | $\sigma(x_1) \cdot (u - l) + l$ | softplus($x_2$) $\cdot s_{\max} + \epsilon$ |
| Dweib | $[0, \infty)$ | $\lvert x_1 + 50 \rvert + \epsilon$ | $\lvert x_2 + 1 \rvert$ |
| Bitwise | $\mathbb{Z}_{\geq 0}$ | Per-bit logits, no activation (BCE on raw logits) |  |
|  | $\mathbb{Z}$ | Signed-magnitude: bit 0 = sign, remaining = $\lvert x \rvert$ |  |
| Poisson | $\mathbb{Z}_{\geq 0}$ | $\lambda = $ softplus($x_1$) $+ \epsilon$ |  |

simplified form: when $\mu < l$, the distribution reduces to a truncated geometric with

$$\log p(n) = \log(1 - \gamma) + (n - l)\log\gamma - \log(1 - \gamma^{u-l+1})$$

and symmetrically when $\mu > u$. Note that this only takes effect when the activation function allows $\mu$ to fall outside of $(l, u)$, or if numerical instability causes small deviations from these bounds.

Furthermore, the denominator of the partition function is clamped to a minimum value of $\epsilon$ before taking the logarithm to avoid $\log(0)$.

### A.5.1 Discrete Weibull implementation

We compute the forward pass of Dweib as follows:

```
1  def forward(alpha, beta, target, eps=1e-12, logeps=-100):
2          """
3          :param alpha: Alpha parameter
4          :param beta: Beta parameter
5          :param target: Target value
6          :param eps:
7          :return:
8          """
9
10         # Calculate terms
11         # We calculate negative powers. For large targets/small alpha, these
12         # can be -inf.
13         term1 = - ((target + eps) / alpha) ** beta
14         term2 = - ((target + 1) / alpha) ** beta
15
16         # Calculate difference
```

```
17            diff = term2 - term1
18
19            # FIX 1: Handle Double Infinity
20            # If both are -inf, diff becomes NaN (inf - inf).
21            # We set it to -inf (prob 0).
22
23            if term1 == −∞ and term2 == −∞:
24                diff = −∞
25
26            # FIX 2: Handle Positive Noise
27            # Diff must be < 0. Floating point can make it > 0 (e.g. 1e-20),
28            # causing log(-expm1(diff)) to be log(negative) -> NaN.
29            if diff > 0:
30                diff = 0
31
32            # FIX 3: Numerical Stability
33            # Use a stable implementation of e^x − 1 for precision with diff near
     0.
34            # - log(1 - exp(diff)) = log( -(exp(diff) - 1) ) = log( -expm1(
     diff) )
35            # - If diff is 0 (clamped), expm1(0)=0, log(0)=-inf (Correct).
36            # - If diff is -inf, expm1(-inf)=-1, -(-1)=1, log(1)=0 (Correct).
37            res = log(- expm1(diff)) + term1
38
39            # Clamp lower bound for stability
40            if res < logeps:
41                res = logeps
42
43            p = exp(res)
44
45            if p < eps:
46                p = eps
47
48            save_for_backward(alpha, beta, target, term1, term2, p, eps)
49
50            return res
```

The backward pass is then:

```
1      def backward(logprobgrad):
2          """
3          :param logprobgrad: Gradient of the loss wrt to the log
     probability
4          :return: Grads for (alpha, beta)
5          """
6
7          alpha, beta, target, term1, term2, p, eps = saved_from_forward()
8
9          den = p + eps
10
11         alpha_num = tet(term2) - tet(term1)
12         alpha_res = (beta/alpha) * (alpha_num/den)
13
14         beta_num = tet(term1) * (target/alpha) \
15                     - tet(term2) * ((target+1)/alpha)
```

```
16            beta_res = beta_num / den
17
18            # Handle NaNs in gradients (usually due to 0/0 in den or infs)
19            if alpha_res is NaN:
20                alpha_res = 0.0
21            if beta_res is NaN:
22                beta_res = 0.0
23
24            return alpha_res * logprobgrad , beta_res * logprobgrad
```

```
1  def tet(x):
2            """
3            Computes x * exp(x), defined as 0 if x = - inf.
4            """
5            if x == −∞:
6                return 0.0
7
8            return x * exp(x)
```

For vectorized versions, see the code under `contin/dweib/dweib.py`.

### A.6   Bitwise training loss weighting

During training, we apply a positional weight of $2^i$ to the binary cross-entropy loss for bit position $i$:

$$\mathcal{L}_{\text{weighted}} = \sum_{i=0}^{k-1} 2^i \cdot \text{BCE}(\hat{x}_i, x_i)$$

This weighting reflects the fact that an error in a higher-order bit has a larger impact on the decoded integer value than an error in a lower-order bit. During evaluation, all bits are weighted equally to obtain the standard log-likelihood.

