# OpenReview forum: "Predicting integers from continuous parameters"
_TMLR — Accepted by TMLR_

### Review · Reviewer_wyPY · 2026-03-17

**Summary Of Contributions:**

This work introduces a series of discrete distributions, parametrized by continuous and learnable parameters, in the context of integer prediction, a problem that is often modeled as a regression one, especially when no bounds over the target are known. The authors’ claim is that it is often hard to interpret the combination of loss terms corresponding to continuous (for integer data) and discrete predictions (for categorical data), whereas using integer-friendly distributions helps to maintain the interpretation. The authors perform an analysis over 6 distribution across multiple datasets, with the goal of understanding which of them works well in practice. The authors also contribute the field by introducing the danorm, dalap and bitwise distributions.

**Audience:**

Yes

**Audience Explanation:**

The ML community has given too little attention to this problem, and it is my belief that this contribution does a great service in testing how we can move beyond the use of categorical and normal distributions in handling discrete labels. There is still too little mixture of neural and probabilistic tools in the literature despite the interesting advantages this may bring.

**Broader Impact Concerns:**

This is a pretty general ML paper, there are no concerns or ethical considerations.

**Claims And Evidence:**

Yes

**Claims Explanation:**

The claims of the paper mostly relate to the effectiveness of the distributions studied and the theoretical contributions. The theory looks reasonable, although I did not take the time to check all passages in detail, whereas the empirical evidence is based on a sound, fair and reproducible setup which has some computational limitation as the authors acknowledged. The evaluation covers a broad spectrum of datasets and use cases which enhances the quality of the analysis.

**Requested Changes:**

- To substantiate more their claims the authors should compare against a Poisson (and a mixture of) as a trivial baseline
- The reason for the previous request is that the Poisson may still work well depending on how the target labels are distributed. The authors are also encouraged to add plots showing the ground truth label distribution (train/val/test) of the different datasets in the appendix.
- The motivation of the paper is not that convincing and should be improved. If I am not mistaken, while the MIDI example corresponds to a use case where the label value is technically unbounded, it could be trivially modeled with a categorical distribution that considers a sufficiently high number of classes. The authors should clarify why this is not a viable solution for most of the use cases (bounded and unbounded) that they consider, and stress more the fact that in the unbounded case a categorical cannot be technically instatiated.
- In this sense, in addition to the Poisson distribution, they could also compare with a categorical or mixture of categorical distributions in addition to the Poisson, where artificial ranges are chosen for the unbounded cases.

---

> ### Author Response · Authors · 2026-04-12
>
> We thank you for your thoughtful review. We have implemented the requested changes as follows:
>
> > To substantiate more their claims the authors should compare against a Poisson (and a mixture of) as a trivial baseline.
>
> We have added the Poisson distribution as a baseline to all experiments except image generation (where it is not applicable).
>
> > The reason for the previous request is that the Poisson may still work well depending on how the target labels are distributed. The authors are also encouraged to add plots showing the ground truth label distribution (train/val/test) of the different datasets in the appendix.
>
> We have added a section to the appendix that provides plots and descriptive statistics for all datasets.
>
> > The motivation of the paper is not that convincing and should be improved. If I am not mistaken, while the MIDI example corresponds to a use case where the label value is technically unbounded, it could be trivially modeled with a categorical distribution that considers a sufficiently high number of classes. The authors should clarify why this is not a viable solution for most of the use cases (bounded and unbounded) that they consider, and stress more the fact that in the unbounded case a categorical cannot be technically instantiated.  In this sense, in addition to the Poisson distribution, they could also compare with a categorical or mixture of categorical distributions in addition to the Poisson, where artificial ranges are chosen for the unbounded cases.
>
> We have fully rewritten the introduction and removed the MIDI example in favor of a simpler motivation. We discuss the categorical distribution and its limitations more explicitly.

---

### Review · Reviewer_iQF1 · 2026-03-27

**Summary Of Contributions:**

This paper studies an interesting and relevant problem: how to model integer-valued targets directly with discrete output distributions whose parameters remain continuous and therefore compatible with gradient-based learning in neural networks. The paper compares several candidate families, including three existing baselines and three additional approaches, namely a continuous-parameter extension of a discrete Laplace analogue (Dalap), a discrete normal analogue (Danorm), and a probabilistic interpretation of bitwise prediction (Bitwise). The empirical study is fairly broad, covering tabular regression tasks, sequential prediction for MIDI timing, and a proof-of-concept image-generation setting. The main takeaway is that Bitwise and Dalap tend to perform best in terms of negative log-loss, while Danorm is often competitive in RMSE, although the paper also acknowledges that, in the tabular setting, a standard continuous relaxation still tends to give the strongest RMSE. Overall, I found the problem formulation and the experimental scope interesting, and I think at least part of the audience would care about the question. However, I remain unconvinced by the paper in its current form because the mathematical development appears insufficiently verified: several derivations in the methods section seem inconsistent, which makes it difficult to fully trust the theoretical claims and, by extension, weakens the strength of the overall contribution.

**Audience:**

Yes

**Audience Explanation:**

I answered **Yes** because the paper addresses a relevant methodological question that I expect would interest at least part of the TMLR audience: how to construct discrete distributions over integer-valued targets whose parameters remain continuous and trainable by gradient-based methods. This is a natural problem in probabilistic deep learning, especially when one wants a discrete predictive model rather than a continuous relaxation. The paper is also reasonably broad in scope, covering tabular, sequential, and image settings, which makes the findings potentially useful beyond a single narrow application. So although I have significant reservations about the current mathematical presentation and the overall level of support for the claims, I do think the underlying question and the comparative findings are of interest to some readers.

**Broader Impact Concerns:**

**I do not have major broader-impact concerns specific to this submission**. This is primarily a methodological paper on discrete probabilistic modeling for integer-valued targets, with experiments on tabular prediction, MIDI timing, and image generation, and I do not see a paper-specific ethical issue that would require a substantial Broader Impact statement beyond standard generic considerations about downstream use of predictive or generative models.

**Claims And Evidence:**

No

**Claims Explanation:**

I answered **No** because, while the paper presents an interesting question and a reasonably broad empirical study, I do not think the current evidence is yet accurate and convincing enough to fully support the strength of the main claims. On the empirical side, the experimental scope is a positive aspect, but the evidence remains somewhat fragile: I did not see uncertainty estimates or variability across runs, and two candidate methods are excluded from the image experiments because of numerical instability or computational impracticality. This makes it harder to assess how robust the observed rankings really are. In addition, some of the conclusions feel slightly stronger than what the numerical margins justify, especially when the differences are small or when important baselines are absent from part of the comparison. So my concern is not that the paper has no evidence, but rather that the evidence is not yet strong enough, either theoretically or empirically, to support the claims at the level of confidence suggested in the writing.

**Mathematical concerns.** My main concern is that several derivations in Sections 3.2, 3.5, and 3.7 appear internally inconsistent, to the point that I am not confident that the stated losses and gradients are correct as written. For the discretized Laplace model, the paper defines the probability mass as a CDF difference over the bin $[n-\tfrac12, n+\tfrac12)$ and the loss as the negative log of that quantity, but the displayed gradients with respect to $\mu$ and $b$ do not seem compatible with differentiating a logarithm of a CDF difference: they contain neither the bin endpoints nor the normalizing denominator $P(n \mid \mu, b)$, and in simple regimes (e.g. $\mu > n+\tfrac12$) they do not match the derivative obtained directly from the stated formula. In Danorm, if $p(n \mid \mu, \gamma) \propto \gamma^{(n-\mu)^2}$, then one should obtain
$$
-\log p(n \mid \mu, \gamma) = \log z - (n-\mu)^2 \log \gamma,
$$
whereas the manuscript writes the opposite sign. Since $\log \gamma < 0$ for $\gamma \in (0,1)$, this is not a cosmetic issue: it changes the interpretation of the loss and would make it decrease as $|n-\mu|$ grows. I also found an apparent algebraic error in the one-sided bounded Dalap normalization: the finite geometric sum should involve a term of the form $1-\gamma^{\lfloor \mu \rfloor + 1}$, but the derivation instead leads to a $\gamma^{\mu+2}$ term, which appears off by one power of $\gamma$. Finally, Section 3.7.1 states that the expectation converges to $\mu$ as $\gamma \to 0$, while Proposition 4 instead states convergence to $r(\mu)$ except in the midpoint case, and the final displayed expectation in the proof seems to reuse the weight $\gamma^{f^2}$ for both neighboring integers. Taken together, these issues make the mathematical presentation feel insufficiently verified at present, and I would need the authors to carefully re-derive and correct these formulas before I could trust the theoretical part of the paper.

**Requested Changes:**

**Requested changes**

1. **Critical for acceptance:** Please carefully re-derive and correct the mathematical development in Sections 3.2, 3.5, and 3.7. As written, several formulas appear inconsistent, including the discretized Laplace gradients, the sign of the Danorm negative log-likelihood, and the bounded Dalap normalization.

2. **Critical for acceptance:** Please verify explicitly that the implemented training objectives match the mathematically correct formulas. If the implementation differs from the manuscript, the paper must be corrected accordingly; if not, the empirical results may need to be revisited.

3. **Critical for acceptance:** Please revise the theoretical claims more conservatively and ensure consistency across statements, propositions, and proofs, especially regarding limiting behavior and expectation formulas.

4. **Important:** Please strengthen the empirical evidence by reporting results across multiple seeds, or at least by providing measures of variability for the main experiments.

5. **Important:** Please tone down or better support the strongest empirical claims, especially in the image setting, where some methods are excluded and some reported differences are very small.

6. **Would strengthen the work:** Please clarify more explicitly when a discrete probabilistic model should be preferred over a continuous relaxation, especially since the latter often remains stronger in RMSE on tabular tasks.

7. **Would strengthen the work:** Please add more implementation details on numerical stabilization, parameterization, and approximation choices, to improve reproducibility.

Overall, the main revision I would require is to make the mathematical core fully reliable and to recalibrate the claims accordingly.

---

> ### Author Response · Authors · 2026-04-12
>
> We thank you for your careful review. We fully agree that the mathematical verification in the original manuscript was insufficient, for which we apologize. We have gone through all derivations and proofs again in detail. We found and fixed various minor errors and rewrote three parts as detailed above.
>
> > I did not see uncertainty estimates or variability across runs, and two candidate methods are excluded from the image experiments because of numerical instability or computational impracticality.
>
> We have repeated runs and added standard errors to all but the image generation experiments. We made substantial efforts to stabilize the implementations of the baselines. We have included our stabilized implementation of the discretized Weibull in the appendix, and added details on the stabilization and activation of other methods as well. While we can not eliminate the possibility that with different techniques the methods that were excluded for instability could still be made to work, we do believe that we have eliminated the possibility that the instabilities are due to naive or overly simplistic implementation. Of course, this is hard to quantify, but we hope the additional implementation details provide some illustration.
>
> > For the discretized Laplace model, the paper defines the probability mass as a CDF difference over the bin and the loss as the negative log of that quantity, but the displayed gradients with respect to and do not seem compatible [...]
>
> These gradients were indeed incorrect. Since we do not use explicit gradients for DLaplace, (the gradients from torch's automatic differentiation are sufficiently stable), and the correct gradients did not add much to the exposition, we have simply removed them.
>
> >  In Danorm, if , then one should obtain whereas the manuscript writes the opposite sign. Since for , this is not a cosmetic issue: it changes the interpretation of the loss and would make it decrease as grows.
>
> There was a minus missing in this formula. To clarify the working of the loss we have written it for both Dalap and Danorm as
>
> $$
> d \log \frac{1}{\gamma} + \log z
> $$
> Where $d$ is the squared or absolute distance between the target and $\mu$.
>
> This shows most clearly that the loss is a linear function of the (squared) distance, with $\log \frac{1}{\gamma}$ a positive value since $\gamma \in (0,1)$.
>
> > I also found an apparent algebraic error in the one-sided bounded Dalap normalization [...].
>
> This was indeed a mistake, which we have fixed. Since this mistake was also present in the code, the Dalap results have changed slightly.
>
> > Finally, Section 3.7.1 states [...]
>
> This mistake has been fixed. The correct expectation is $r(\mu)$. We rewrote the proof to make it more precise.
>
> We have implemented all requested changes. Specifically:
>
> > Critical for acceptance: Please carefully re-derive and correct the mathematical development in Sections 3.2, 3.5, and 3.7. As written, several formulas appear inconsistent, including the discretized Laplace gradients, the sign of the Danorm negative log-likelihood, and the bounded Dalap normalization.
>
> Described above.
>
> > Critical for acceptance: Please verify explicitly that the implemented training objectives match the mathematically correct formulas. If the implementation differs from the manuscript, the paper must be corrected accordingly; if not, the empirical results may need to be revisited.
>
> We verified all implementations. In the case of the one-way bounded Dalap, and unbounded Bitwise this required us to rerun some experiments, due to minor mistakes.
>
> > Critical for acceptance: Please revise the theoretical claims more conservatively and ensure consistency across statements, propositions, and proofs, especially regarding limiting behavior and expectation formulas.
> > Important: Please tone down or better support the strongest empirical claims, especially in the image setting [...].
>
> We have rewritten the results and discussion to make the claims more conservative.
>
> > Important: Please strengthen the empirical evidence by reporting results across multiple seeds, or at least by providing measures of variability for the main experiments.
>
> Implemented, except for the image generation where this was infeasible. The results or the MIDI experiment are still running.
>
> > Would strengthen the work: Please clarify more explicitly when a discrete probabilistic model should be preferred over a continuous relaxation [...].
>
> We have rewritten the introduction to highlight the conceptual benefits of discrete output distributions. So far, our results do not give a clear answer for when a discrete distribution provides a performance benefit with respect to a continuous relaxation. We leave this for future work.
>
> > Would strengthen the work: Please add more implementation details on numerical stabilization, parameterization, and approximation choices, to improve reproducibility.
>
> We have added these to the appendix.

---

### Review · Reviewer_sqUF · 2026-03-27

**Summary Of Contributions:**

This paper considers machine learning tasks where we wish to use a neural network to make predictions over integer-valued labels, for example MIDI notes (0-127), pixels (0-255), or counting data. One example of the techniques evaluated here is to use a neural network to produce a single real value which is used to define the mean of a (discretized and truncated) normal distribution. Our model's prediction over the integers is then this distribution, and we can compute the log loss, for example.

The paper contains lots of discussion and calculations for various ways to perform such transformations. It then gives some experimental results comparing to when these approaches are better than other approaches.

I had trouble understanding the background information as presented in the introduction. The running example is MIDI music generation. I did not understand the discussion of drawbacks for the approach presented: even if it's unclear how to combine loss functions for tone and duration in a principled way, can't I use a holdout set to tune their relative weights? Even worse, the submission does not discuss the fact that modern neural networks are exceedingly good at generating music: a transformer can have categorical outputs and will learn the relationships between them.

Overall, while there is some interesting investigation going on, the submission failed to convince me that these techniques represent a meaningful improvement over existing approaches.

**Audience:**

No

**Audience Explanation:**

As discussed above, the current version of the paper does not present a sufficient argument that its techniques are of interest for the problems it considers.

It's definitely possible that the core ideas could be turned into a publication which is of general interest, but this would require major changes.

**Broader Impact Concerns:**

None.

**Claims And Evidence:**

Yes

**Claims Explanation:**

Update: after revision, I am satisfied that the paper's claims are accurate and supported by evidence.

---------------------------

As discussed above, I failed to understand much of the presentation around why these techniques are helpful. I suspect the submission is missing a level of abstraction: it starts with an informal discussion and then moves directly to manipulating specific distributions. The story seems incomplete without a more technical overview of how these distributions will actually be applied.

The experiments by themselves are not enough to convince me of the paper's main claims. The MIDI experiments are performed using an LSTM and the image generation experiments are based on PixelCNN++ (Salimans, 2017). The fact that the paper can improve on such baselines is only weak evidence that "traditional" approaches are limited, since the state of the art has moved so dramatically in the past decade.

The same applies to the tabular count data: I do not understand how or why the approaches here could improve on existing techniques for predicting social media upvotes, for example.

**Requested Changes:**

This paper needs substantial revisions before I would be comfortable accepting it. As a starting point, I would need a much clearer description of a setting where current techniques fail (or are inadequate).

---

> ### Author Response · Authors · 2026-04-12
>
> We thank you for your time and for your thoughtful comments.
>
> > I had trouble understanding the background information as presented in the introduction. The running example is MIDI music generation. I did not understand the discussion of drawbacks for the approach presented: even if it's unclear how to combine loss functions for tone and duration in a principled way, can't I use a holdout set to tune their relative weights? Even worse, the submission does not discuss the fact that modern neural networks are exceedingly good at generating music: a transformer can have categorical outputs and will learn the relationships between them.
>
> We believe the original introduction may have given an incorrect impression of our motivations. We used MIDI generation as a motivating example, since that is how we came to this problem, but it is only meant to be an illustration of one case where integer output distributions may be useful: when combining losses over multiple discrete output values. We have now motivated this as follows in our rewritten  introduction:
>
> > This conceptual elegance has practical benefits. Consider a setting where we must predict multiple values, say a flight simulator where we choose our engine thrust from an integer range and our flight stick position from the unordered set $\{\text{up}, \text{down}, \text{left}, \text{right}, \text{neutral}\}$. Modeling both values as probabilities results in log-losses that can be neatly and simply summed together. While it's not guaranteed that the losses will balance, adding them together without a multiplier gives us a reasonable place to start, since the magnitude of the two losses is proportional to the representational complexity of the two values. That is, we can describe the current state of the controls in $- \log p(\text{thrust}) - \log p(\text{stick})$ bits, so summing the two log-losses together is a natural way to balance them.
> > Compare this with the case where we model the thrust with a continuous relaxation. In that case we get a value of arbitrary magnitude for the thrust's log-density, which could even be negative. When we add the log probability density to the log-probability for the stick position, we are forced to add a multiplier, which must be tuned. Moreover the sum of the two losses is no longer an interpretable value.
>
> In short, continuous outputs can indeed be tuned on held out data, but this costs both data and time. Because of the coding interpretation, discrete log probabilities are much more likely to balance when added directly without balancing parameters.
>
> > The experiments by themselves are not enough to convince me of the paper's main claims. [...] The fact that the paper can improve on such baselines is only weak evidence that "traditional" approaches are limited, since the state of the art has moved so dramatically in the past decade.
>
> We would like to clarify here that our aim is explicitly not to advance on the state of the art in either music or image generation. This is outside of our scope and indeed beyond the resources we have available. The idea behind the experimental design is to compare different output distributions on a level playing field. For this purpose, we fix the upstream architecture, and vary only the output distributions. The use of an RNN is purely practical: it is likely powerful enough to do well on the task, and removes modeling choices like context length and tokenization from the experiment design. Choices that we would need to tune and justify with a more modern transformer-based approach.
>
> In the case of image generation, we explicitly did not choose diffusion models (the current state of the art). We have added the following explanation to the paper (abridged here due to the character limit).
>
> > The current state of the art in this domain is almost exclusively built on diffusion models. However, diffusion is not the most natural place to test our integer models. In (Ho et al 2020)  for example, a separate decoder module is required in the last generation step to map the continuous representation of the image to an integer representation. Even then, the authors note
> > [...]
> > Since we are interested in using compression (that is, the negative log loss) as our main metric of evaluation---with perceptual quality a secondary consideration---we will focus on likelihood-based generative models, specifically pixel-wise auto-regressive models.
>
> We emphasize again that the fact that Dalap is competitive with the original PixelCNN++ cannot be explained simply from the fact that PixelCNN++ is an older model, because our method uses _the same model_. We simply change the output distribution and keep the rest of the model the same, including most hyperparameters.
>
> We consider this a more precise evaluation of the output distribution than simple raw FID or BPD performance across all upstream model variations, which is far out of scope for one TMLR paper.

---

> ### Author Response · Authors · 2026-04-12
>
> > The same applies to the tabular count data: I do not understand how or why the approaches here could improve on existing techniques for predicting social media upvotes, for example.
>
> The tabular datasets were chosen for having realistic integer targets, not for whether their underlying tasks were interesting as use cases. We hope the rewritten introduction provides some illustration for how these distributions might improve existing methods in areas with discrete predictands.
>
> > As a starting point, I would need a much clearer description of a setting where current techniques fail (or are inadequate).
>
> Our claim is not that a discrete output distribution will uniformly improve performance, or that current techniques fail. We merely claim that  using an output distribution that better suits the target data can make things conceptually more simple, and limit the complexity of the model configuration and hyperparameter space.
>
> We believe that this is what makes the paper interesting for the TMLR audience. We do not present it as a method that will always make models perform better, but rather as an additional tool in the modeller's toolkit. One which may make models more elegant and simple in certain settings, by more directly modeling the structure of the target label space. While simplicity and elegance are subjective notions, we hope that the revised text better explains how this may translate to tangible benefits in terms of performance, less hyperparameter tuning and better model comparability.

---

> > ### Comment · Reviewer_sqUF · 2026-05-28
> >
> > I thank the authors for their responses and updates. They improve the paper and in particular it is easier for the reader to understand the claims being made.
> >
> > The paper still does not sufficiently motivate its solutions. The new introduction lays out many drawbacks of other approaches (e.g., neural networks with categorical output must learn ordinality; it may be unclear how to combine loss functions; one may want to measure bits, not RMSE). But it is not clear to me that, taken together, these drawbacks establish that there is a hole in the literature that this work fills.

---

### Author Response · Authors · 2026-04-12

We have uploaded a substantially revised version of the manuscript. We will respond to each review in detail below. Here is an overview of the most substantial changes.

* We have re-written the introduction and removed the motivating example of MIDI music. This example still remains as one of the tasks we evaluate on, but we motivate the research more generally from different angles.
We have checked every mathematical derivation and proof in the paper. This has led to various minor corrections, and to three more substantial changes:
   * The derivatives of the discretized Laplace were incorrect. Since the correct derivatives are complicated, and can easily be computed by automatic differentiation, we have simply removed the derivatives.
    * The section that described behavior of the integer regularization term $log (\gamma^c + \gamma^f)$ was incorrect, since the regularization actually works in the opposite direction. We explain the true effect of this term in detail. Since this does not add much to the understanding of Dalap (but does explain a counter-intuitive aspect of the functional form of the loss), we moved this section to the appendix.
    * We rewrote the proof of Proposition 4 to be more precise in its application of limits.
* We repeated all experiments 10 times with different seeds and report standard errors, except for the image generation task, where this is not feasible. This is finished for the tabular data and included in the revision. We expect the results for the MIDI task by the end of the week (17th of April).
* We included the Poisson distribution as a baseline in the tabular and MIDI tasks.
* We added a section to the appendix which provides descriptive statistics and target value histograms for all datasets.
* We rewrote the results and discussion sections to remove any claims that are not borne out by the evaluation.

---

### Author Response · Authors · 2026-04-24
**Official Comment by Authors**

We are writing to provide a quick update that a new version of our manuscript has just been uploaded.
As mentioned in our detailed responses a few days ago, this latest revision includes the updated results table for the MIDI dataset.
Thank you again for your constructive feedback and for your patience while we completed these runs.

---

### Author Response · Authors · 2026-04-29
**Official Comment by Authors**

We have just uploaded a minor update to the manuscript. We realized that the revision we uploaded a few days ago (which included our finalized MIDI results) inadvertently contained an older working draft of our Conclusion and Limitations sections.

We have now updated these final two sections to ensure they accurately summarize and reflect the empirical findings already presented in the Results section.

Please note that this is the only edit made in this upload. No experimental data, results, tables, or other sections of the text have been altered from our previous revision.

We sincerely apologize for any confusion this oversight may have caused, and we thank you again for your continued time and constructive feedback.

---

### Author Response · Authors · 2026-06-29
**Official Comment by Authors**

We thank the reviewer and the action editor for the careful final check and for the constructive guidance throughout the revision process. We have uploaded a revised camera-ready version nd explicitly checked the remaining consistency issues raised in the final comments. We have also done a full final pass through the document to check for any remaining issues.

In particular, we corrected the Danorm negative log-probability formulation in Section 3.7, in agreement with reviewer iQF1, and made several additional mathematical and textual consistency fixes in Sections 3.3, 3.5, 3.7, Appendix A.3, and Appendix A.5. We also corrected the MAESTRO sample counts, clarified the MIDI encoding description, clarified the mixture-model setup, updated the image FID table and discussion, removed duplicated text about excluded PixelCNN distributions, updated the OpenReview URL, and made minor wording, grammar, and formatting fixes throughout.

We appreciate the positive assessment and the detailed final review.

---

### Decision · Action_Editor_quyZ · 2026-05-28

**Recommendation:** Accept with minor revision

**Additional Comments:**

Final comment from Reviewer `iQF1`:

> I thank the authors for the substantial revision and for their detailed response. The manuscript appears to have improved considerably: the motivation is clearer, the empirical evidence has been strengthened through additional baselines and repeated runs, and the added implementation details make the work easier to assess. Most of my initial concerns therefore seem to have been addressed. I would nevertheless ask the authors and the action editor to verify that the final uploaded version is fully consistent with the corrections described in the response.

> In particular, in the PDF version currently available to me, Section 3.7 still seems to contain the old sign issue in the Danorm negative log-likelihood: for $p(n \mid \mu, \gamma) \propto \gamma^{(n-\mu)^2}$, the negative log-probability should be written as $(n-\mu)^2 \log(1/\gamma) + \log z$, not with the opposite sign. Since the authors state in their response that this has been corrected, this may simply be a versioning issue, but I would like the final manuscript to be checked carefully for this and for any remaining textual inconsistencies before acceptance. Overall, my assessment is positive, conditional on this final consistency check.

**Audience:**

Yes

**Audience Explanation:**

The paper addresses a relevant methodological question for parts of the TMLR audience: how to model integer-valued targets using discrete output distributions with continuous trainable parameters. This is relevant to probabilistic deep learning, neural prediction with structured discrete labels, and likelihood-based modeling. The revised paper also provides a comparative empirical study across tabular, sequential, and image-generation settings. Although one reviewer remains unconvinced that the paper fills a major gap in the literature, the other reviewers view the problem as interesting and insufficiently studied, and the work seems likely to be useful to at least some readers.

**Claims And Evidence:**

Yes

**Claims Explanation:**

- The revised manuscript appears to support its main claims
- The authors reworked the motivation, corrected mathematical derivations, verified the implementation, reran affected experiments where necessary, added additional baselines, added repeated runs and standard errors for the main non-image experiments, and toned down the empirical claims.
- While one reviewer still raises a concern about the final consistency of the uploaded version, this appears to be a minor revision/check issue rather than a fundamental objection to the soundness of the contribution.